



# Scour variability across offshore wind farms (OWFs): Understanding site-specific scour drivers as a step towards assessing potential impacts on the marine environment

Karen Garcia[1*,] Christian Jordan[1], Gregor Melling[3], Alexander Schendel[1,2], Mario Welzel[1], Torsten Schlurmann[1,2]

[1]Leibniz University Hannover, Ludwig-Franzius-Institute for Hydraulic, Estuarine and Coastal Engineering, Nienburger Str. 4, 30167 Hannover, Germany. Email: schendel@lufi.uni-hannover.de, jordan@lufi.uni-hannover.de, welzel@lufi.uni-hannover.de, schlurmann@lufi.uni-hannover.de

[2]Coastal Research Centre, Leibniz University Hannover & Technical University Braunschweig, Merkurstraße 11, 30419 Hannover, Germany.

[3] Federal Waterways Engineering and Research Institute (BAW), Wedeler Landstraße 157, 22559 Hamburg, Germany. Email: gregor.melling@baw.de

[*]Corresponding author: garcia@lufi.uni-hannover.de

**Abstract:** The development of offshore wind farms (OWFs) is critical to meeting renewable energy targets, but predicting scour around offshore wind energy structures (OWES) and the associated potential impacts on marine ecosystems remains a challenge. Using high-resolution bathymetry data, this study analyses field-measured scour depths at 460 monopiles at nine British OWFs. The analysis reveals a large spatial variability of relative scour depths ($S/D$) between OWF sites, but also within individual wind farms. Principal component analysis (PCA) is used to identify significant drivers of this variability. When the entire data set is considered, results indicate that median grain size ($D_{50}$), relative water depths ($h/D$), and the significant wave height ($H_{s,99}$) are the most important influencing factors for the variability of scour depths. Other parameters investigated, such as Froude number ($Fr$), pile Reynolds number ($Re$), flow intensity ($U_{c,99}/U_{cr}$), and current velocity ($U_{c,99}$), were found to have a less clear influence. Further sediment-specific analysis shows that relative water depth ($h/D$) is a particularly relevant driver of scour at sites with fine (63 to 200 $\mu m$) and medium sands (200 to 630 $\mu m$), with larger scour depths occurring in shallower water depths. Findings from this study provide new insights into scour behavior across a range of spatial and environmental scales and lay a foundation for the transferability of scour prediction frameworks to new OWF sites. In the future, findings and datasets from this study are suggested to be used to estimate scour-induced sediment transport and thereby to provide a step towards the assessment of potential impacts of OWF expansion scenarios in the marine environment. By addressing the broader implications for regional sediment dynamics, this research contributes to the sustainable development of offshore wind energy.

Keywords: Offshore wind farms (OWFs), scour depths, monopile, sediment transport, principal component analysis (PCA).



## 1 Introduction & Motivation

The expansion of renewable energy is crucial for a sustainable and independent energy supply. In order to meet the
European Union's targets for expanding offshore wind energy (EU, 2020), it is necessary to develop areas with
previously unveiled metoceanic and geophysical conditions. To this end, existing knowledge gaps about the interaction
of individual offshore wind energy structures (OWES) or entire offshore wind farms (OWFs) with the marine
environment must be closed. In general, the disturbance of the flow by an offshore structure causes scour, which might
not only affect the structure's stability (Saathoff et al., 2024), but the mobilized sediment may also contribute to the
overall regional sediment transport (Vanhellemont et al. 2014; Baeye et al. and Fettweis et al., 2015; Rivier et al. 2016)
with potential impacts on the marine environment.
The scour process itself, is a multivariate process, which is dependent on a combination of complex hydrodynamic and
geotechnical drivers. Early studies focused on the understanding of the scour process around a pile under simplified
isolated hydraulic conditions, such as steady flow (e.g., Sheppard et al., 2004; Zhao et al., 2012; Sarkar et al., 2014;
Baykal et al., 2015), unsteady and bidirectional tidal currents (e.g. Escarameia and May1999; McGovern et al., 2014;
Yao et al., 2016; Schendel et al, 2018) and waves (e.g. Sumer et al.,1992b; Carreiras et al., 2001; Stahlmann et al.,
2013). With the availability of more sophisticated experimental facilities and numerical models, research is
increasingly shifting toward more complex hydrodynamic loads consisting of a combination of waves and currents, as
in the studies of Sumer and Fredsøe (2001), Qi and Gao (2014), Schendel et al (2020), Lyu et al. (2021), and Du et al.
(2022) but also towards studies addressing complex offshore structures (Welzel, 2021; Welzel et al., 2024; Sarmiento
et al., 2024; Chen et al. 2025).
Despite those advances in scour research, uncertainties remain in current scour prediction methods (Chen et al., 2024).
Matutano et al. (2013) demonstrated the challenges of applying empirical formulas for maximum scour depths by
comparing different methods with data from ten European OWFs, revealing overpredictions in all but two cases. The
comparison highlights the fundamental challenge of accounting for complex marine flow conditions, characterized by
the superposition of multiple influencing factors, such as flow velocity, sediment coarseness, and wave-current
interactions, in the prediction of scour processes using existing models (Gazi et al., 2020; Harris eat al. 2023)
Compared to laboratory experiments focusing on scour processes, rather few studies are based on in-situ data, that
represent the actual scour development under complex flow conditions. These studies assessed the scour at individual
structures, such as monopiles (Walker, 1995; Noormets et al., 2003; Harris et al., 2004; Rudolph et al., 2004;
Louwersheimer et al., 2009), and jackets (Bolle et al., 2012; Baelus et al., 2018; Harris and Whitehouse et al., 2021),
or dealt with larger datasets from entire offshore wind projects (DECC 2008; COWRIE 2010; Whitehouse et al. 2010;
Whitehouse et al., 2011; Melling (2015)), covering both spatial and temporal evolution of scour under different
hydrodynamic regimes and seabed types across the North Sea and British continental shelf. In general, the amount and
variety of field data collected has increased with the gradual installation of offshore wind turbines. Focusing
specifically on the correlation between scour and on-site conditions, Melling (2015) analyzed the relationships between
the variations of scour hole dimension within OWFs and both sedimentological and hydrodynamic parameters of 281
turbines in the Outer Thames estuary. Melling's (2015) study, although only covering three OWFs, represents one of
the most comprehensive investigations of field related scour to date, with the highest number of structures examined



so far. By comparing field data with physical modeling experiments and literature, the study provided valuable insights
into the range of observed scour and its controlling structural hydrodynamic, and sedimentological parameters.
In addition to local scour at individual structures, the cumulative effect of multiple structures in an OWFs can alter
ocean dynamics (Christiansen et al., 2022), mixing (Schultze et al., 2020) and sediment mobility (Vanhellemont &
Ruddick, 2014). Increased velocities and turbulence induced by OWFs has also the potential to affect the marine
environment, potentially leading to global erosion around the structures as well as habitat loss or gain for benthic flora
and fauna (Shields et al., 2011; Wilson and Elliott, 2009; Welzel et al., 2019). . Concerns over the potential impacts of
OWF installations on local ecosystems further include collision risks, noise pollution, electromagnetic field and the
introduction of invasive species (Lloret et al., 2022; Bailey et al., 2014; Teilmann and Carstensen, 2012; Watson et al.,
2024). As the size and scale of OWF increases, the risk of significant cumulative effects arising is also expected to
increase (Brignon et al., 2022; Gușatu et al., 2021). The drivers and interdependencies of these large-scale processes
are not yet well understood and the precise impact of scour induced sediment transport on the marine environment
remains uncertain, highlighting the need for interdisciplinary research utilizing field data.
In order to gain a better understanding of the geophysical changes following the installation of OWFs and potential
impacts on the marine environment arising from it, this study analyses the scour development at OWES as a first step.
This study builds its analysis on field data, including high-resolution bathymetry scans from British OWFs, which have
recently been made publicly available. This provides an opportunity to extend the understanding of scour evolution
and its key drivers using a cross-regional dataset. A total of 460 monopiles were analyzed to obtain local scour depths
and their spatial distribution in dependence of selected hydrodynamic and geological drivers.
Understanding scour development is a critical first step in assessing potential environmental impacts. It will help
determine whether OWES and entire OWFs contribute to regional sediment mobilization and provide a foundation for
future research into the long-term morphological footprint of OWF installations and their broader ecological effects.
Towards the overall goal, the paper focuses on advancing understanding of scour at OWES by analyzing field data
from 460 monopiles across 9 OWFs, situated in diverse ocean regimes with current velocities from 0.54 m/s to 1.77
m/s (99th quantile), significant wave heights from 1.5 m to 2.7 m (99th quantile), water depths from 5 to 35 m and
grain sizes ranging from cohesive sediment (51.54 $um$) to medium gravel (19872 $um$). The spatial distribution and
variability of scour depths across and within these OWFs are determined and correlated with selected hydrodynamic
and sedimentological parameters, using Principal Component Analysis (PCA). Ultimately, the results of the study will
help decrease uncertainty in scour depth prediction by assessing the contribution of the main drivers of scour
development from multivariable field data.
This paper is organized as follows: Section 2 describes the study area and methodology in which the methods used to
obtain the scour depths and selected on-site parameters are explained in detail (subsections 2.2 – 2.5). Additionally,
the application of the Principal Component Analysis (PCA) to identify the primary correlation between these
parameters and scour development is explained (subsection 2.6).  The results are presented in section 3, followed by
implications for scour predictions for OWF (section 4), limitations and future research (section 5) and ending with the
conclusions (section 6).





## 2 Study area and methodology

### 2.1 Study area

The research area, located in British waters, is illustrated in Figure 1, showing the specific locations of the nine studied OWFs. Figure 1A provides a general overview, while Figure 1B pinpoints the positions of the OWFs, labeled 1 to 9. These OWFs correspond to Robin Rigg, Barrow, Teesside, Humber Gateway, Lincs, Lynn and Inner Dowsing, Greater Gabbard, London Array, and Gunfleet Sands OWFs, respectively. Figures 1C and 1D display the 99th quantiles of the significant wave heights ($H_s$) and current velocity magnitudes ($U_c$) at the nine locations, respectively.

Notably, wind farms such as Robin Rigg and Barrow are situated in the Irish Sea, while the remaining seven are located in the North Sea at the east coast of UK (Fig 1B). Water depths ($h$) ranging from 5 to 35 m can be found across the nine OWFs. Depth data ($h$) were obtained from EMODET (**https://emodnet.ec.europa.eu/en/bathymetry**). The OWF located in the shallowest water depths is Robin Rigg with $h$ ranging from 1 to 14 m (Fig. 1B). Conversely, the OWF with the deepest water depths is Greater Gabbard with $h$ ranging from 21 to 35 m (Fig. 1B).

The highest and lowest significant wave heights (99th quantile) can be found at Humber Gateway OWF ($H_s$ = 2.7 m) (Fig. 1C-D) and at Gunfleet Sands OWF ($H_s$ = 1.5 m), which are located at the mouths of the Humber and Thames estuaries (Fig. 1C-D), respectively. Regarding the quantile of current velocities, the highest value is found at Robin Rigg OWF with 1.8 m/s (Fig. 1C-D), while the lowest value is found at Gunfleet Sands OWF with a value of 0.4 m/s (Fig. 1C-D).

Depending on the locations of the OWFs, the seabed conditions vary from sandbanks featuring a variety of bedforms to intertidal mudflats. Accordingly, the sediment also varies from silt to coarse and very coarse gravel, with the sediment at Teeside OWF consisting of fine and silty sands and that at Humber Gateway consisting of sandy gravel and boulders. In contrast, OWFs such as London Array and Greater Gabbard are located in the Outer Thames Estuary with sandbanks and channels, while others such as Barrow and Robin Rigg have distinct geological features such as megaripples, mudflats and deposits from different geological eras.

### 2.2 Data description

Bathymetric datasets from the nine OWFs considered in this study were collected via multibeam echosounder (MBES) before, during and after the construction of the OWFs and were afterwards made available by its operators via the Marine Data Exchange (MDE).

In total 460 OWES (of 680 available) with monopiles foundations were analyzed in this study. For the correlation between scour and hydrodynamic conditions at the nine studied OWFs, metocean hindcast datasets (i.e., significant wave height ($H_s$) and velocity magnitude ($U_c$)) by the Copernicus Marine Service (CMEMS) (https://marine.copernicus.eu/) were used (CMEMS, 2023a, 2023b).



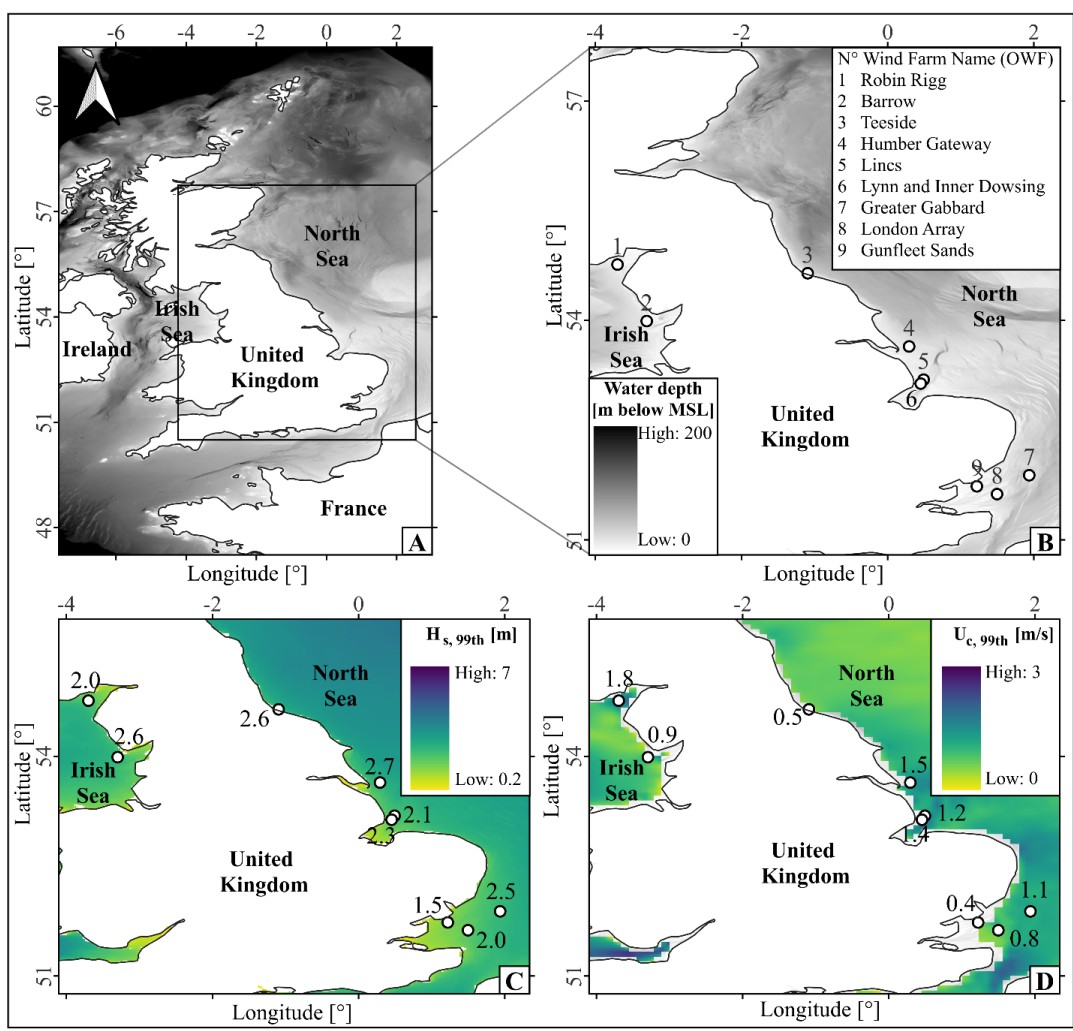

**Figure 1: A) Study area. B) Location of the nine studied OWFs. Shown bathymetry data originates from EMODET (https://emodnet.ec.europa.eu/en/bathymetry). C) 99th quantile of significant wave heights ($H_s$) based on data for the year 2012. D) 99th quantile of current velocity magnitudes ($U_c$) based on data for the year 2012.**





Table 1 shows the OWFs considered in this study and provides an overview of their structural characteristics as well
as the hydrodynamic and geotechnical site conditions. Pile diameters ($D$) were obtained from Negro et al. (2017), water
depths ($h$) are based on EMODET (2020), $D_{50}$ represents the median grain diameter of the sediment. The sediment
data shown in Table 1 were obtained in Phi units from each OWF's benthic reports, then converted to $D_{50}$ values in
micrometers ($\mu m$) according to Bunte et al. (2001). The scour depth $S$ represents the deepest scour at an individual
OWES. The number of turbines varies from 26 turbines installed at Teesside OWF to 174 turbines installed at London
Array OWF, indicating the different operational scales. For some OWFs, including Lynn and Inner Dowsing, extensive
bathymetric data spanning over ten years was available. In contrast, others, such as Humber Gateway, had more limited
bathymetric data with a coverage duration of four years. The highest grid resolutions of the bathymetric datasets found
at each OWF varied from 0.2 to 0.5 m, with the highest resolution of the bathymetries found at each OWF being used.
The earliest bathymetry was collected at Barrow OWF in 2005 and the most recent was collected at Lynn and Inner
Dowsing in 2017, highlighting the long-term monitoring efforts at the wind farms. However, in this study only scour
depths obtained from the pre- and the first post-construction bathymetries were considered.  The shortest period
between pre and post bathymetries was found at Lincs OWF with 377 days between August 2010 and August 2011,
while the longest period between scans was detected at Greater Gabbard OWF with 2902 days (~8 yrs) between June
2005 and May 2013.
Furthermore, environmental and hydrodynamic conditions associated with each OWF are also shown in Table
1, which are essential for understanding how different variables contribute to scour around monopiles. These variables
include the 99[th] quantile significant wave height ($H_s$), representing the average height of the highest third of waves.
The wave height has a direct influence on the wave-induced current velocity near the seabed and thus strongly
determines the bed shear stresses and the formation of the vortex system around the OWES (Sumer & Fredsøe, 2002;
Schendel et al., 2018). The 99[th] quantile current velocity magnitude ($U_c$) indicates the resultant of eastward and
northward tidal flow components, whereas $U_{crit}$ depicts the critical flow velocity for sediment entrainment. Their ratio,
the flow intensity ($U_c/U_{cr}$), is a key parameter in describing the general sediment mobility and has a large impact on
the scour rate and depth (Melville and Coleman, 2000). The relative water depth ($h/D$) influences the formation of the
horseshoe vortex in such a way that the size of the horseshoe vortex is reduced as the flow depth decreases, resulting
in a reduction in the scour depth. At greater water depths ($h/D \geq 5$) the scour depth becomes almost independent of
water depth (Sumer and Fredsøe, 2002).
The Froude number ($Fr$) and pile Reynolds number ($Re$) are used to characterize the flow conditions around
the pile. The Froude number indicates whether the flow is dominated by gravitational or inertial forces. With increasing
Froude number, pressure gradients at the pile increase, which affects the flow field in the vicinity of the pile and
typically leads to larger scour depth. The Reynolds numbers provides information on whether the flow is laminar or
turbulent, and determines the characteristics of the vortex system around the pile.
Dimensionless parameters as given in Table 1 were calculated based on the equations summarized in Table

187 2.







| OWF name | N° of OWES | Pile diameter $D$ (m) | | Scour depths $S$ (m) | Water depths $h$ (m) | $D_{50}$ ($\mu m$) | Flow | | | | | | | |
| | | | | | | | Wave height $H_{s,99}$ (m) | Current Velocity $U_{c,99}$ (m/s) | Critical velocity $U_{cr}$ (m/s) | Relative scour depths $S/D$ | Relative water depths $h/D$ | Froude number $Fr$ | Reynolds number $Re$ | Flow intensity $U_{c,99}/U_{cr}$ |
|---|---|---|---|---|---|---|---|---|---|---|---|---|---|---|
| **Robin Rigg** | 60 | 4.3 | Min | 1.3 | 5 | 167 | 2.36 | 1.55 | 0.39 | 0.30 | 1.03 | 0.13 | $5.14 \times 10^6$ | 3.51 |
| | | | Max | 10 | 14 | 267 | 2.59 | 1.77 | 0.44 | 2.32 | 3.07 | 0.23 | $5.86 \times 10^6$ | 4.43 |
| **Barrow** | 30 | 4.75 | Min | 0.98 | 15 | 138 | 2.43 | 0.91 | 0.46 | 0.20 | 3.67 | 0.06 | $3.50 \times 10^6$ | 1.89 |
| | | | Max | 6 | 23 | 445 | 2.52 | 1.11 | 0.48 | 1.20 | 4.71 | 0.08 | $4.26 \times 10^6$ | 2.40 |
| **Teesside** | 26 | 5 | Min | 0.65 | 8 | 51 | 2.52 | 0.54 | 0.42 | 0.13 | 2.08 | 0.04 | $2.10 \times 10^6$ | 1.19 |
| | | | Max | 1.62 | 20 | 166 | 2.76 | 0.54 | 0.45 | 0.32 | 3.49 | 0.05 | $2.10 \times 10^6$ | 1.29 |
| **Humber Gateway** | 72 | 4.2 | Min | 0.5 | 15 | 5918 | 2.24 | 1.51 | 1.53 | 0.11 | 3.65 | 0.11 | $4.87 \times 10^6$ | 0.58 |
| | | | Max | 2.51 | 20 | 19000 | 2.37 | 1.56 | 2.62 | 0.59 | 4.65 | 0.12 | $5.06 \times 10^6$ | 0.99 |
| **Lincs** | 75 | 5.2 | Min | 0.54 | 12 | 505 | 2.47 | 1.07 | 0.50 | 0.10 | 2.41 | 0.08 | $4.29 \times 10^6$ | 1.31 |
| | | | Max | 1.92 | 21 | 1982 | 2.71 | 1.67 | 0.86 | 0.38 | 3.88 | 0.13 | $6.71 \times 10^6$ | 3.12 |
| **Lynn and Inner Dowsing** | 60 | 4.74 | Min | 0.5 | 9 | 684 | 2.11 | 1.30 | 0.51 | 0.10 | 2.10 | 0.11 | $4.76 \times 10^6$ | 1.63 |
| | | | Max | 2.35 | 17 | 1950 | 2.36 | 1.45 | 0.80 | 0.49 | 3.47 | 0.13 | $5.29 \times 10^6$ | 2.53 |
| **Greater Gabbard** | 139 | 6 | Min | 0.5 | 23 | 394 | 2.41 | 1.02 | 0.51 | 0.08 | 3.50 | 0.05 | $4.72 \times 10^6$ | 1.14 |
| | | | Max | 4.54 | 35 | 2296 | 2.67 | 1.22 | 1.00 | 0.75 | 5.83 | 0.07 | $5.64 \times 10^6$ | 2.25 |
| **London Array** | 174 | 7 | Min | 1.2 | 1 | 120 | 1.89 | 0.71 | 0.32 | 0.21 | 0.31 | 0.04 | $2.56 \times 10^6$ | 1.14 |
| | | | Max | 9.5 | 27 | 930 | 2.36 | 0.81 | 0.61 | 2.02 | 4.67 | 0.19 | $3.56 \times 10^6$ | 2.33 |
| **Gundfleet Sands** | 49 | 4.7 | Min | 0.88 | 2 | 146 | 1.52 | 0.48 | 0.34 | 0.18 | 0.54 | 0.03 | $1.74 \times 10^6$ | 1.05 |
| | | | Max | 7.73 | 16 | 253 | 1.72 | 0.86 | 0.45 | 1.64 | 3.34 | 0.09 | $3.12 \times 10^6$ | 2.07 |

**Table 1. Overview of studied OWFs with hydrodynamic and sedimentological site conditions.**






**Table 2.** Calculation of the variables included in the analysis

| Variable | Equation | |
|---|---|---|
| Current velocity | $U_{c,99} = \sqrt{u_0^2 + v_0^2}$ | (1) |
| Froude number | $Fr = \dfrac{U_c}{\sqrt{gh}}$ | (2) |
| Pile Reynolds number | $Re = \dfrac{U_{c,99}D}{v}$ | (3) |
| Relative density | $s = \dfrac{\rho_s}{\rho_w}$ | (4) |
| Relative grain size | $D_* = \left(\dfrac{\rho g}{v^2}\right)^{\frac{1}{3}} D_{50}$ | (5) |
| Critical Shields | $\theta_{cr} = \dfrac{0.3}{1 + 1.2D_*} + 0.55(1 - \exp(-0.02D_*))$ | (6) |
| $U_{cr}$ | $U_{cr} = 7 * \left(\dfrac{h}{D_{50}}\right)^{\frac{1}{7}} (g(s-1)D_{50}\theta_{cr})^{0.5}$ | (7) |
| Flow intensity | $\dfrac{U_{c,99}}{U_{cr}}$ | (8) |


Sediment density ($\rho_s$) is 2650 kg/m³, a value assumed for all OWFs sites based on Soulsby (1997). The water density
($\rho_w$), which represents sea water at the surface, is 1027 kg/m³. The kinematic viscosity ($v$) is 1.3e-6 m/s², and the
gravitational acceleration (g) is 9.8 m/s². Equations 4 and 6 are taken from Soulsby and Whitehouse (1997), where $s$
represents the specific gravity of sediment grains. Equation 5 was calculated based on van Rijn (1984), where $D_*$ is the
non-dimensional grain diameter, and this is first calculated to calculate the critical Shields parameter ($\theta_{cr}$), which
corresponds to the initiation of motion at the bed. The northward ($v_0$) and eastward ($u_0$) current velocities in the
water column in a temporal resolution of 1 hour are used to calculate the $U_{c,99}$.

In addition, the minimum and maximum values of the variables at each OWF are shown. The 99[th] quantile of
significant wave heights ($H_{s,99}$) in a temporal resolution of 3 hours and current velocities ($U_{c,99}$) in a time window of
1 hour are used in this study. The 99[th] quantile was chosen for this study, due to scour development being more driven
by largest $H_{s,99}$ and $U_{c,99}$. The datasets were obtained between pre- and post- construction bathymetries. The data were
collected over a one-year period, prior to the post-construction bathymetry.





Since some hydrodynamic parameters (e.g. wave period or orbital flow velocity) and geotechnical data (e.g. grain
density and size distribution) were not directly available for all locations during the preparation of this study, it was
not possible to determine other dimensionless parameters (Shields, KC, Ucw) that are important for the scour process
with sufficient reliability.

**2.3 Pre-processing of bathymetric data**

Figure 2 shows the workflow used in this study, starting with the acquisition of bathymetric datasets, originally
obtained from the Marine Data Exchange, and their conversion to Ordnance Datum Newlyn (ODN). This was followed
by the generation of 100m x 100m tiles for each available bathymetric dataset, centered on each turbine location. If
bathymetric scans with different spatial resolutions were available for the same date, only the one with the highest
resolution was used. In addition, some turbine locations could not be further analysed due to missing pre-construction
scans or poor data quality. Tiles with more than 50% empty cells were discarded because a high percentage of missing
data increases the likelihood that important areas, such as the scour region, are poorly captured. Tests were conducted
with lower missing cell thresholds (10% and 25%), but even with 50% missing data, valuable information for scour
analysis was retained. Using a stricter 25% threshold, too many tiles were lost, including those that still contained
useful data. As a result, 460 of the 680 turbines in the nine OWFs were analyzed in more detail.
The difference in bed elevation at turbine sites between the pre-construction (Fig 2.A) and post-construction surveys
(Fig 2.B), was used for extracting scour information. The deepest scour at each turbine site was then extracted from
the difference plot (Figure 2.C). A detailed description of this part of the workflow is provided in the next chapter.






**Figure 2: General workflow and methodology used to assess the scour distribution and evolution as well as the**
**correlation between scour parameters and site conditions. A) Pre-installation scan. B) Post-installation scan. C)**
**Difference plot after subtraction of B from –A. D) Map of spatial distribution of scour depths. E) Principal**
**Component Analysis (PCA). F) Site conditions of wave heights and current velocities.**





### 2.4 Calculation of scour parameters

First, to eliminate outliers, a threshold based on the 99[th] percentile was used to filter out extreme values, ensuring that outliers did not skew subsequent analyses or visualizations. Subsequently, to address potential offsets between pre- and post-construction, a median filter was applied to both datasets. The difference in medians, excluding the presumed scour area, was considered the offset. This offset was then applied while calculating the difference plot between the pre- and post-construction bathymetries (Fig. 2A-C). To remove additional outliers close to the turbine, an area equivalent to 110% of the pile's foodprint area was excluded from the center of the difference plot.

The deepest scour depth (see green dot in Fig. 2C) was then extracted from the difference plot (Fig. 2C). The calculated scour depths were then visualized to show the spatial distribution across the nine OWFs (Fig. 2D).

### 2.5 Principal component analysis (PCA)

In the case of field data, the correlation of the scour process with hydrodynamic and geotechnical variables is complicated by the simultaneous change of several of these variables. In order to reduce the complexity and simplify this multivariate problem, PCA was used in a next step (Fig. 2.E). PCA works by transforming the data into a set of new variables called principal components, which are combinations of the original variables (Jolliffe & Cadima 2016). These components are ordered based on how much variance they explain, with the first principal component (PC1) explaining the maximum variance in the data, followed by the second principal component (PC2). Each component also has an eigenvalue, which shows the amount of variation it captures. Generally, the PCA is able to handle lots of independent variables and helps to simplify the data without losing important information (Harasti, 2022). Unlike studies that use PCA for variable reduction (Harasti, 2022), in our analysis we retained all principal components to identify and quantify the relationships between the selected variables.

In this study, the PCA was applied to a dataset of 692 turbines, including 177 turbines from London Array OWF and 100 turbines from Thanet OWF, based on Melling's (2015) data. The PCA was then performed using eight independent variables that contributed to the principal components. Those variables were the relative water depths ($h/D$), wave height ($H_{s,99}$), current velocity ($U_{c,99}$), Reynolds number ($Re$), Froude number ($Fr$), sediment size ($D_{50}$), flow intensity ($U_{c,99}/U_{cr}$), and the relative scour depths ($S/D$).

In order to correlate the relative scour depths ($S/D$) with the other variables, the PCA biplot was employed to estimate the correlation between the variables (Gabriel et. al., 1971). In the biplot, the angle between the respective vectors indicates the degree of correlation, with an angle close to 0° indicating a strong positive correlation, while 180° indicates a strong negative correlation and 90° indicates no correlation at all. Each correlation percentage was thus calculated by taking the cosine of the measured angle between vectors, thereby providing an estimation of how closely variables within the analysed dataset are related the relative scour depths (S/D).

An additional approach to reducing the complexity of multivariate datasets is to initially group the data based on a selected key variable. Accordingly, the PCA was also applied to the dataset after it had been grouped by grain size ($D_{50}$ diameter) classes (Annad et al., 2021), given that the sediment characteristics of the seabed play a significant role in local scour (Qi et al., 2016). This approach facilitated a more precise estimation of local scour, thereby reducing uncertainties related to sediment.



## 3 Results

### 3.1 Spatial distribution of scour depths

To illustrate the variability in scour depths between the nine studied OWFs and within single OWFs, Figure 3 shows the spatial distribution of scour depths. There are clear differences between OWFs in both the magnitude and variability of scour depths. For example, at OWF Robin Rigg (Figure 3.A), the highest scour depths were identified, the values range from 0.29 $S/D$ to 2.49 $S/D$. This OWF is characterized by fine and medium sands. In contrast, the smallest scour depths occurred at the OWF of Lincs and Lynn and Inner Dowsing (Figure 3.E and 3.F), with values from 0.12 $S/D$ to 0.92 $S/D$, which is possibly linked to coarse sands presented at both sites . Furthermore, the highest variability ($\sigma = 0.44$) in scour depths were detected at OWF London Array (Figure 3.H) and Barrow (Figure 3.B), likely influence by the complex seabed morphologies and sediment compositions in these areas. On the other hand, the significant variability at London Array may be explained by the presence of the Long Sand and Kentish Knock sandbank. This illustrates how different site characteristics can result in various scour distributions, even within a single OWF.

The remaining OWFs showed relatively low scour depths and little spatial variability, even though site conditions were significantly different, as indicated by their seabed conditions from very fine sand for Teesside (Figure 3.C) to coarse and very coarse gravel for Humber Gateway (Figure 3. D).



**Figure 3:** Spatial distribution of relative scour depths ($S/D$) at the nine studied OWFs. Numbered markers (1-9) denote the locations of Robin Rigg, Barrow, Teesside, Humber Gateway, Lincs, Lynn and Inner Dowsing, Greater Gabbard, London Array, and Gunfleet Sands OWFs, respectively. The upper colormap represents water depths, with darker shades indicating deeper water. The lower colormap indicates relative scour depths, with darker blue color indicating largest scour. Black filled squares represent turbines with scour protection,

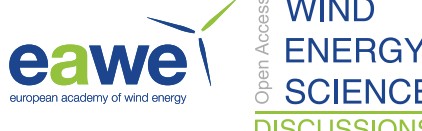

**while empty circles denote missing data. Shown bathymetry data originates from EMODET**
**(⊙).https://emodnet.ec.europa.eu/en/bathymetry).**

## 3.2 Principal component analysis (PCA)

The analysis of Figure 3 reveals notable variations in scour depths across individual OWFs. This variance underscores
the need for a more detailed examination of specific wind farm characteristics to identify the drivers of scour. To this
end, a PCA was conducted to correlate scour depths and selected parameters by identifying and quantifying their
relationships. The PCA biplot presented in Figure 4 illustrates these correlations between scour depths and the studied
variables and provides a comprehensive view of how different factors interact and influence scour depths.

a)

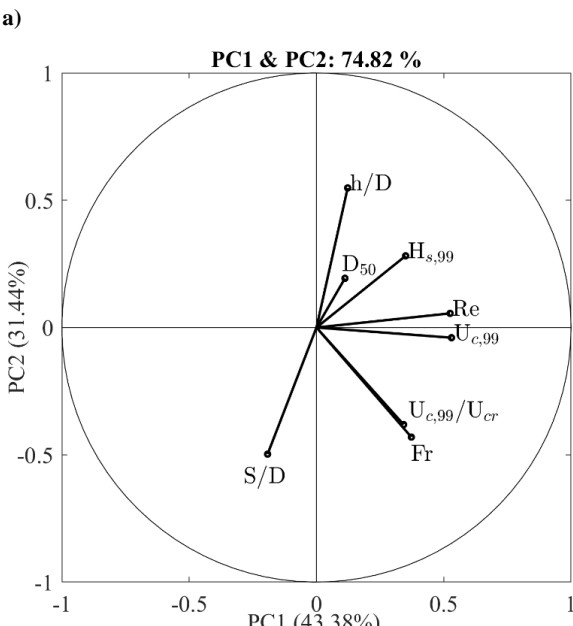

b)

| Variables | $\theta$ to S/D | % Correlation with S/D |
|---|---|---|
| $D_{50}$ | **171.49** | **0.989** |
| $h/D$ | **171.053** | **0.988** |
| $H_{s,99}$ | **149.91** | **0.865** |
| $Fr$ | 61.9 | 0.469 |
| $Re$ | 117.2 | 0.456 |
| $U_{c,99}/U_{cr}$ | 63.0 | 0.453 |
| $U_{c,99}$ | 106.8 | 0.289 |

**Figure 4: a) PCA biplot, illustrating the correlation between variables and relative scour depths ($S/D$). b) The**
**table detailing the angles between $S/D$ and the other variables, as well as their corresponding percentage**
**correlation.**
As shown in the biplot, PC1 and PC2 account for 74.82% of the variation in the data set. This high percentage indicates
that these two components capture most of the significant patterns in the data, allowing for a meaningful interpretation
of the relationships among the variables. In the biplot, each vector stands for a variable, with the direction and
magnitude of the vector reflecting its contribution to the principal components. The variables that contribute the most
to the variance in PC1 are flow velocity ($U_{c,99}$) and Reynolds number ($Re$), with loadings of 0.5303 and 0.5248,
respectively. In contrast, the variance in PC2 is primarily explained by the relative water depths ($h/D$) and the relative
scour depths ($S/D$), with loadings of 0.548 and 0.49732, respectively. This significant contribution of flow velocity
($U_{c,99}$) and Reynolds number ($Re$) to PC1 suggests that variations in these hydrodynamic parameters are critical in
shaping the principal dynamics of the dataset.





The table (Fig. 4b) next to the biplot provides further insight by showing the angular distances between the $S/D$ vector
and each of the other variables, as well as their respective correlation coefficients. One of the key observations is that
despite its relatively small contribution to the total variance (as indicated by the shorter vector length in the biplot),
sediment size ($D_{50}$) has the strongest negative correlation with scour depths ($S/D$), as indicated by a correlation
coefficient of 0.989. This highlights the critical influence of sediment size on scour processes, even though it does not
account for much of the variance captured by the first two principal components.
This observation can be explained by the underlying physical processes that affect scour depths. As noted by
Whitehouse (2010) for non-cohesive sediments, larger sediment sizes are more resistant to erosion, resulting in reduced
scour depths. Therefore, while $D_{50}$ is strongly correlated with scour depths, it does not explain the broader variability
in the data that is influenced by other factors. Variables such as flow velocity ($U_{c,99}$), Reynolds number ($Re$), and
Froude number ($Fr$), although less correlated with scour depths, contribute more to the total variance. This suggests
that these flow related variables influence scour depths through more complex or non-linear interactions with other
hydrodynamic conditions and sediment characteristics.
Given that the initial PCA analysis indicates the strongest negative correlation between $D_{50}$ and $S/D$, a more in-depths
investigation of the influence of $D_{50}$ on scour processes is required. Since the sediment characteristic of the seabed
plays a significant role to local scour (Qi et al. 2016), the PCA was applied to the same data set, but pre-clustered into
different soil classes (Annad et al. 2021). By reducing the uncertainties related to sediment size, this analysis should
provide a better estimation of the local scour. This classification also facilitates the identification of parameters that
are more influential in estimating scour for specific soil classes rather than uniformly across different types.
**3.3 Principal component analysis (PCA) by clustered soil classes**



WIND
ENERGY
SCIENCE
DISCUSSIONS

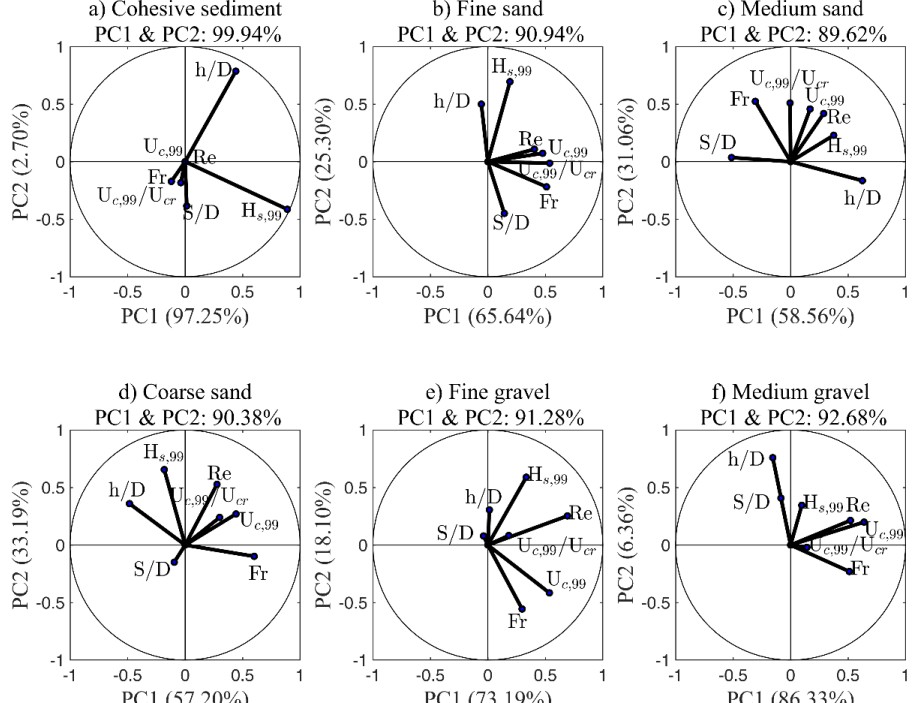

**Figure 5: PCA correlation by clustered grain size classes between the remaining 7 dimensionless parameters, including the scour depth. a) Cohesive sediment ($D_{50}$ ≤63 $\mu m$). b) Fine sand (63 to 200 $\mu m$). c) Medium sand (200 to 630 $\mu m$). d) Coarse sand (630 to 2000 $\mu m$). e) Fine gravel (2000 to 6300 $\mu m$). f) Medium gravel (6300 to 20000 $\mu m$). Clustering of the grain size ($D_{50}$) was based on Annad et. al. (2021).**

Building on the initial PCA analysis, which emphasized the significant influence of grain size ($D_{50}$) on scour depths ($S/D$), a more detailed investigation was conducted by categorizing the dataset into six grain size classes: cohesive sediment ($D_{50}$ ≤63 $\mu m$) with 5 data points, fine sand (63 to 200 $\mu m$) with 203 data points, medium sand (200 to 630 $\mu m$) with 206 data points, coarse sand (630 to 2000 $\mu m$) with 221 data points, fine gravel (2000 to 6300 $\mu m$) with 19 data points, and medium gravel (6300 to 20000 $\mu m$) with 73 data points.

Figure 5 shows PCA biplots for each soil class illustrating the relationships between scour depths ($S/D$) and the variables $h/D$, $H_{s,99}$, $U_{c,99}$, $Fr$, $Re$, and $U_{c,99}/U_{cr}$. The first two principal components (PC1 and PC2) explain between 90.98% and 99.55% of the variance within each class, thus describing more of the variance in comparison to when the PCA was applied to all data. Data complexity seems to be greatly reduced by just removing the effect of sediment. In the cohesive sediment class (Figure 5a), PC1 dominates, explaining the majority of the variance, suggesting a primary underlying pattern that drives the variability in scour depths. However, this result must be interpreted with caution as





the analysis in this group is based on only 5 data points. In contrast, the fine sand (Figure 5b) and medium sand (Figure
5c) classes show a more balanced contribution from PC1 and PC2.
Analysis of the correlations between scour depths ($S/D$) and other variables within each soil class reveals different
patterns. In the cohesive sediment class ($D_{50} \leq 63\ \mu m$), relative scour depths is positively correlated with flow intensity
($U_{c,99}/U_{cr}$). This suggests that as flow intensity increases, scour depths tends to increase, which meets physical
expectations in clear water conditions, i.e. stronger flow intensity leading to larger scour (Melville, 2008). However,
the data points in this cluster belong to the Teeside OWF, for which flow intensities ($U_{c,99}/U_{cr}$) between 1.17 and 1.28
m/s were determined and thus live-bed flow conditions. In contrast to clear-water conditions, the scour depths under
live-bed conditions are influenced by the migration of bed forms, typically leading to scour depths smaller than that in
clear-water conditions. It is important to note that the flow intensity vector ($U_{c,99}/U_{cr}$) is remarkably short, reflecting
its minimal contribution to the overall variability, despite its positive correlation with scour depths ($S/D$). Although
$U_{c,99}/U_{cr}$ is positively correlated with scour depths, its limited impact on the total variance captured by PC1 and PC2
suggests that other factors may have a stronger influence in this class.
In contrast, relative water depths ($h/D$) has a strong negative correlation with scour depths in fine sand (63 to 200
$\mu m$) and medium sand (200 to 630 $\mu m$). This indicates that as relative water depths increases, scour depths tends to
decrease in these finer sediment classes. From a physical view, Melling (2015) found out that in similar substrates,
scour depths agree well between different geographic locations. Furthermore, Melling (2015) showed that turbines
located in sandy sediments seemed to show a strong influence of relative water depths on scour, insinuating that
geotechnical factors are less influential in granular sediments. The decrease in scour depths with water depth seems
unexpected, as in shallow water a greater water depth should lead to a larger boundary layer and thus potentially
stronger horseshoe vortex and scour depths (Melville, 2008). However, as explained by Harris and Whitehouse (2014),
a weaker down flow and hence a weaker horseshoe vortex can be expected in deeper water. In deeper water, the
hydrostatic component of the total energy at the front of a pile becomes more significant compared to the kinetic
component, resulting in a more uniform pressure field on the upstream side of the pile and a stagnation point closer to
the seabed. In addition, the thinner boundary layer implied by shallower relative water depths could consequently also
lead to greater bed shear stresses, resulting in generally greater sediment mobility.
The dynamics observed in coarse sand (630 to 2000 $\mu m$) and fine gravel (2000 to 6300 $\mu m$) are different from the
finer sediments.
In these classes, the Reynolds number ($Re$) and the Froude number ($Fr$) show significant negative correlations with
scour depth, indicating that higher values of these parameters correspond to reduced scour depth. Again, these trends
are again somewhat unexpected. In coarser sediments the formation and migration of bed forms is limited, which
should results in a stronger correlation with flow parameters and scour depths.
For medium gravel (6300 to 20000 $\mu m$), water depth has a positive correlation with scour depth, meaning that greater
relative water depths are associated with greater scour depths in coarser sediments. The data points in the cluster can

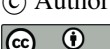



be attributed to the Humber Gateway OWF, which is the only OWF that features clear-water conditions. Given the
large grain sizes, a smaller influence of flow parameters on the variability of scour depths should be expected

**3.4 Correlation of scour depth with main drivers**

Following the PCA (Figure 5), which identified the primary variables influencing scour depths ($S/D$) across sediment
classes, a Pearson correlation analysis was performed to quantify the strength and direction of these relationships.
Figure 6 shows the Pearson correlation results for each cluster and the variable with the strongest correlation, with the
red lines representing the linear regression fit and the correlation coefficients shown in red text. The Pearson correlation
was calculated by the following equation:
$$R = \frac{\sum(x_i-\bar{x})(y_i-\bar{y})}{\sqrt{\sum(x_i-\bar{x})^2\,\sum(y_i-\bar{y})}} \dots\dots\dots\dots\dots (9)$$

Taking into account the small number of data points in this sediment cluster, scour depths at locations with cohesive
sediments (Fig. 6a) show a moderate correlation between scour with flow intensity ($U_{c,99}/U_{cr}$). For the fine and
medium sand clusters, the PCA revealed a similarly strong dependence of scour depth on relative water depth ($h/D$).
Plotting scour depths against relative water depths now shows a clearer trend and hence dependence for the medium
sand sites (Fig. 6c) than for the fine sand sites (Fig. 6b). The Pearson coefficients of -0.56 and -0.86 confirm this
difference in the dependence of scour depth on relative water depth. The correlations of the fine and medium sand
clusters are supported by a larger number of data points, increasing the reliability of the findings.

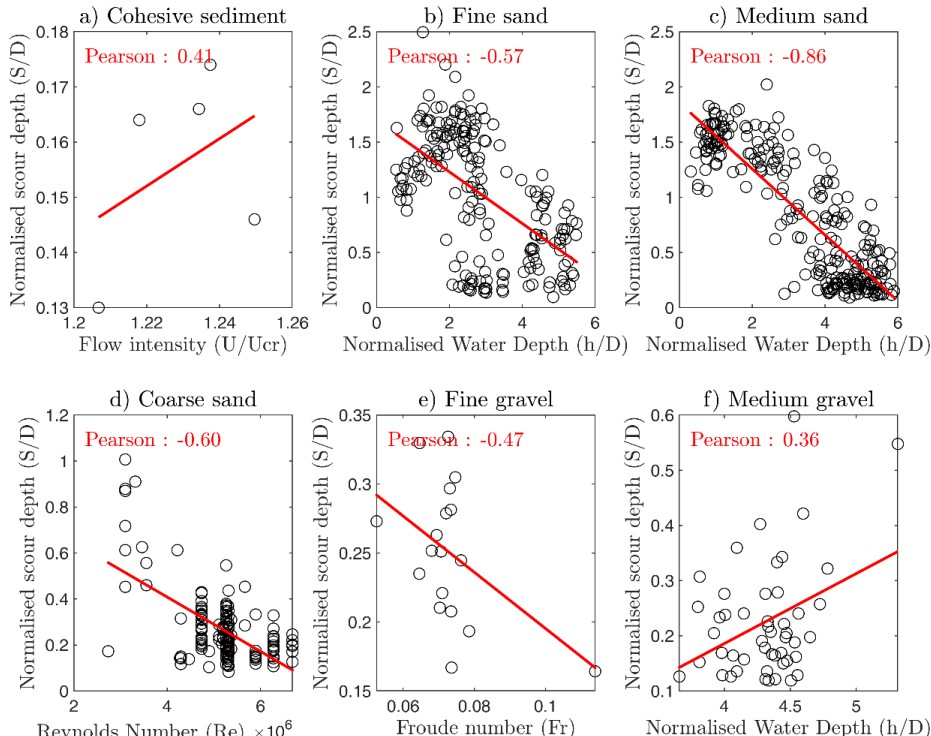

**Figure 6: Pearson correlation of representative variables obtained by PCA analysis with scour depths across different grain sizes classes.** **a) Cohesive sediment ($D_{50} \leq 63 \; \mu m$). b) Fine sand (63 to 200 $\mu m$). c) Medium sand (200 to 630 $\mu m$). d) Coarse sand (630 to 2000 $\mu m$). e) Fine gravel (2000 to 6300 $\mu m$). f) Medium gravel (6300 to 20000 $\mu m$).**

For the coarse sand (630 to 2000 $\mu m$), the PCA analysis revealed the strongest correlations between scour depth and pile Reynolds number. The comparison of these two parameters in Figure 6d confirms the general trend that higher Reynolds numbers lead to lower scour depths, but also shows that scour depths can vary considerably within an OWF despite identical Reynolds numbers. Due to the grid size of the flow data used, the current velocities and hence the pile Reynolds numbers vary only slightly within an OWF.

For fine gravel (2000 to 6300 $\mu m$), the PCA analysis showed a very strong correlation between scour depth and Froude number. However, it is difficult to derive this trend between depth and Froude number from the comparison of these two parameters in Figure 6e. Rather, the Froude numbers for the few data points in this group of grain sizes are very close to each other, rendering the correlation unreliable.

Finally, medium gravel (6300 to 20000 $\mu m$) displays a positive correlation between scour depth and water depth, with a Pearson coefficient of 0.36. This indicates that larger relative water depths correspond to increased scour depth,





although the range of this increment remains small (between 0.1 and 0.4 S/D). This variation in scour depth is relatively
minor compared to the trends observed in fine and medium sands, where changes in water depth yield more pronounced
differences in scour depth. The smaller impact in medium gravel  may be attributed to the generally greater resistance
of larger sediments to scour, even with increasing water depth.
The most significant trends emerge from the fine sand (63 to 200 $\mu m$) and medium sand (200 to 630 $\mu m$), where strong
negative correlations between relative scour depths and relative water depths are observed. This suggests that
significant scour occurs in shallower waters with finer sediments. Such findings highlight the importance of relative
water depths as a key factor influencing scour processes in specific sediment types, emphasizing that scour
management and predictions for offshore structures should take sediment characteristics and relative water depths into
account. These results are consistent with the studies from Melling (2015) and Harris and Whitehouse (2014), which
also show a decrease in scour depths in finer sediments as water depth increases. This negative correlation can be
explained by the reduction in bed shear stress with increasing water depth, which limits sediment mobilization,
particularly in fine and medium sands (Sumer & Fredsøe, 2002; Fredsøe & Sumer, 2014). However, those results are
not in agreement with experimental work where scour around a monopile weakens with reducing relative water depths
(e.g. May and Willoughby, 1990; Whitehouse, 1998). Consequently, relative water depths is included as a parameter
in many empirical formulas, especially in for scour around bridge piles with limited water depth (eg. Laursen, 1963;
Hancu, 1971; Breusers et al., 1977; May and Willoughby, 1990; Richardson et al., 2001). Besides that, these insights
from field data are critical for the accurate assessment and planning of offshore infrastructure installations, particularly
in regions with varying sediment characteristics.








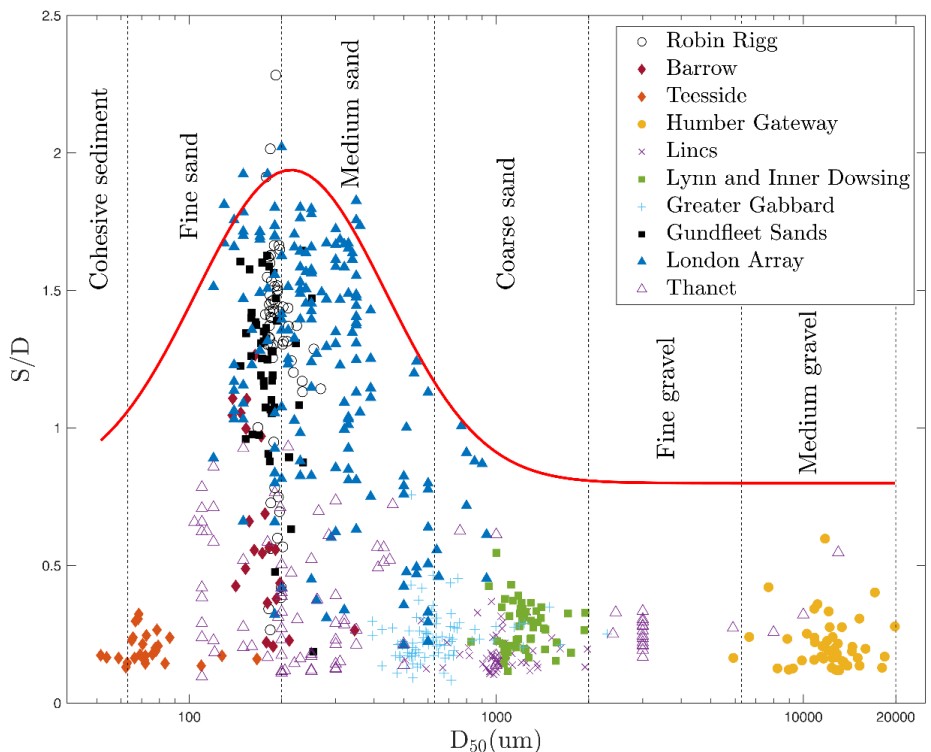


**Figure 7: Relative scour depths S/D against grain size D$_{50}$. Red line gives approximate upper limit of S/D for various D$_{50}$. Data points for London Array and Thanet OWFs are included from Melling (2015).**

Figure 7 summarizes the findings from the PCA analysis (Figure 4) by plotting the relationship between the relative scour depth ($S/D$) and grain size $D_{50}$ across all the sampled locations. This figure illustrates a discernible trend where the largest scour depths scour occur predominantly in fine to medium sands, as indicated by the Gaussian fit line which approximates the upper limit of $S/D$ for various $D_{50}$. This visualization captures the broad distribution of data points and highlights the significant influence of grain size on scour depths, confirming the results PCA of the PCA that identified $D_{50}$ as a key factor in scour dynamics. The trend shown in Fig. 7 is well explained. In general, the mobility potential of the sediments decreases with increasing grain size, which leads to lower scour depths for coarser sediments. Very fine sediments, on the other hand, are subject to the influence of cohesion forces that reduce their erodibility, which also leads to lower scour depths. Therefore, fine and medium sandy sediments have the largest scour potential, which is reflected in the data of Fig. 7. The different symbols represent the OWF, highlighting the geographic spread and variability within the dataset. However, it is important to note that the majority of the data points fall within the range of fine to medium sands, potentially skewing the interpretation.





452 .

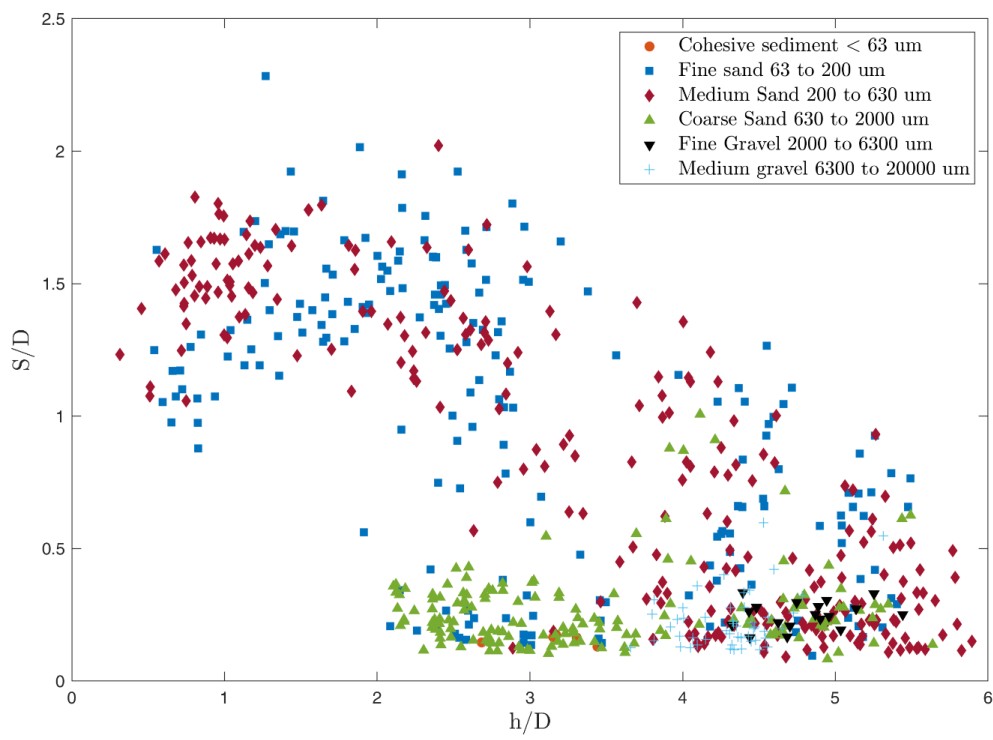

453

**Figure 8: Relative scour depths vs relative water depths, and sediment classification. Data points for London Array and Thanet OWFs are included from Melling (2015).**

In addition to the influence of sediment grain size, relative water depth has been shown to be the most important factor influencing relative scour depth. However, it should be noted that water depth has a direct effect on other parameters. For example, not only is the Froude number formed with water depth, but water depth also significantly determines the potential influence of waves on the development of scour, which in this study has so far only been considered via the significant wave height. It is therefore not clear whether the influence of water depth on scour depth is a causal factor, or whether the cause of changes in scour depth is related to changes in flow conditions caused by changes in water depth. Nevertheless, Figure 8 illustrates the comprehensive correlation between the relative scour depth ($S/D$) and the relative water depth ($h/D$), with the differently colored points representing the studied sediment clusters.

The trend observed figures 6b and 6c is reaffirmed in Figure 8. A distinct relationship exists between the scour depth and water depth in these two sediment types, i.e. both fine sand (63 to 200 $\mu m$) and medium sand (200 to 630 $\mu m$) show that the scour depth decreases with increasing water depth. This trend appearing throughout the bigger dataset emphasizes strong negative correlation between water depth and scour depth for those sediment classes. This behavior



is consistent with findings from previous analyses that identified water depth as a critical factor in shaping scour
dynamics (Whitehouse et al., 2010 and Melling et al., 2015).
In contrast, for sediments with median grain diameters above coarse sands ($D_{50} \geq 630\ \mu m$) the scour depth remains
relatively constant and shows little variability. Figure 8 suggests a generally stable relationship between scour depth
and water depth for these sediment classes, where changes in water depth do not significantly alter scour depth.
However, there are a few exceptions. For example, some locations with coarse sand located in deeper water exhibit
unexpectedly large scour depths. These outliers might stem from site-specific conditions such as dynamic sandbanks
and highly variable, as seen at the London Array OWF (Sturt et al., 2009). These unique environments, characterized
by flow recirculation and sediment mobility, can lead to deviations from expected scour behavior (Melling et al., 2015).
The results for fine and medium sands suggest a potential influence of water depth in reducing scour depth, which
could have implications for sediment transport and the marine environment. Although these results are preliminary,
they provide a first step in understanding how offshore wind turbines could affect sediment redistribution in regions
dominated by these sediment types and small water depth.

**3.5 Detailed analysis of scour patterns for selected OWFs**
Following the observed overall trend shown in Figure 8, this section moves on to examine scour patterns within
individual OWFs, such as Robin Rigg, Lynn and Inner Dowsing, and London Array. This specific analysis will assess
whether the global relationship between scour depths, $D_{50}$, and relative water depths holds under the unique
environmental conditions of each site. This section aims to further our understanding of the dynamics between
sediment characteristics and scour processes by a detailed analysis of the variation within each wind farm to determine
if these global correlations are consistent at the local scale or if there are deviations due to site-specific factors.
**3.5.1 Robin Rigg OWF**
Robin Rigg is presented and discussed in this section as this OWF has the largest overall scour depths of all the OWFs.
This detailed analysis will help to investigate whether the negative correlation between S/D and h/D observed globally
in Figure 8 holds true under variable geotechnical conditions, taking into account that sediment grain sizes range from
fine to medium sands.
Figure 9 shows the distribution of scour depths at Robin Rigg in relation to the variable geotechnical and hydrodynamic
site conditions. This sequence begins with Figure 9A, showing the spatial distribution of scours measured one year
after turbine installation. A significant variation in scour depths in different areas of the OWF can be observed, with
the deeper scour depths mainly located in the northeastern part, particularly around turbines D7, C6, B5 and B4, which
are located in the shallowest waters. Figure 9B shows the spatial distribution of the median grain diameter $D_{50}$ in the
uppermost sediment layer in 2005, with sediment sizes predominantly in the range of fine to middle sand (182 µm to
268 µm). Turbines in areas with finer sands, such as D4, D5, and D6, are observed to generally experience the large
scour, consistent with previous observations by Whitehouse (2006) that finer sand substrates are more susceptible to
scour.

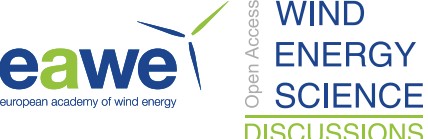

Figure 9C shows the correlation of $S/D$ and $h/D$, classified by colored points which represent sediment grain size
from figure 9B. Contrary to the clear negative correlation between $S/D$ and $h/D$ observed globally in Figure 8, Figure
9C shows a wide distribution of data points with no clear trend, suggesting that local factors in addition to relative
water depths and sediment type have an influence on scour at this site.
For additional insight, Figures 9D and 9E show the distribution of the directions of significant wave heights, as well
as the directions of current velocity magnitudes one-year period, prior the post scan. The highest wave heights came
predominantly from the southwest, which should influence sediment mobility and thus scour structures along this
direction and especially in shallow water depths where wave-induced shear stresses should be higher. Similarly, the
tidal current, with its main directions of south-west and north-east, should result in a change in scour depths along this
main axis. However, a clear trend of scour depth changing in this direction is not given for Robin Rigg.

513 .



**Figure 9: A) Spatial distribution of relative scour depths ($S/D$) from 2008-2009 at Robin Rigg OWF. B) Grain-size distribution. C) Relative scour depths vs relative water depths, and grain size classification D) Significant wave heights E) Current velocities.**

This comprehensive analysis using Figures 9A to 9E shows that while trends obtained from global findings provide a useful baseline for understanding scour, the actual scour observed at Robin Rigg does not necessarily follow those trends. While the distribution of scour depths appears to be strongly influenced by local environmental conditions such



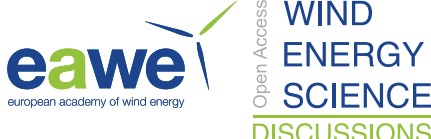

as sediment type, waves and currents, the dominant influence among these cannot be clearly identified, rather the
distribution of scour depths appears to be due to the interaction of all influences.
The discrepancies between the local scour behavior at Robin Rigg and the broader trends observed in Figure 8
underscore the need for site-specific assessments. Such detailed analyses are critical to the development of effective
scour management and mitigation strategies tailored to the unique conditions of each offshore wind farm.

### 526 3.5.2 Lynn and Inner Dowsing OWF

Lynn and Inner Dowsing was chosen as a further example as this OWF had the lowest scour depths of all the OWFs
investigated and is also characterized by coarse to very coarse sands.  Figure 10 provides the same analysis as Figure
9 by providing insight into how local conditions compare to the global trend seen in Figure 8. Figure 10A shows the
spatial distribution of relative scour depths (S/D) measured from 2007 to 2010. Figure 10A shows that the largest scour
depths are mainly concentrated in the Inner Dowsing area, especially around turbines ID1, ID2, ID8, ID9, ID12, ID24,
and ID30. Except for turbine L21, which has the deepest scour depths in the entire wind farm and which is located at
the southeastern end. The significant scour depths observed at certain locations (e.g., D30, L21) are related to cable
exposure (EGS Ltd, 2012; EGS Ltd, 2013), while smaller scour depths are more common in the southern region.
Overall, the spatial distribution shows a slight trend of increasing scour depths from south to north.



**Figure 10: A) Spatial distribution of relative scour depths ($S/D$) at Lynn and Inner Dowsing OWF from 2007-2010. B) Grain-size distribution. C) Relative scour depths vs relative water depths, and grain size classification. D) Significant wave heights E) Current velocities.**





Continuing with the spatial overview, Figure 10B introduces the spatial distribution of $D_{50}$ median grain sizes, which
shows a range from coarse to very coarse sands (695 to 1951 $\mu m$).. The correlation between relative scour depth ($S/D$)
and relative water depth ($h/D$) is examined in Figure 10C.  Similar to Robin Rigg, this OWF does not display the
negative correlation as seen globally in Figure 8, suggesting that additional local factors may significantly influence
scour depths.
Consequently, the significant wave heights and current velocities from hindcast data are shown in Figure 10D and 10E.
The highest wave heights, observed from the northeast, and strong tidal currents flowing from southwest to northeast,
highlight the dynamic environmental forces at play. The presence of the largest scour depths in the Inner Dowsing area
align with the direction of the highest tidal current velocities (Fig. 10E) recorded in the northeast part as well the main
direction of waves. Therefore, the direction of both tidal current and waves likely play a significant role for the scour
development in this wind farms, as the seabed conditions and water depth locally do not exhibit a distinct correlation.

### 3.5.3 London Array OWF

Following the previous results, the analysis for London Array OWF shows a wide range of scour depths from 0.2 $S/D$
to 2.1 $S/D$. This variability differs markedly from the consistently larger scour depths observed at Robin Rigg and the
limited maximum depths of up to 1.0 $S/D$ at Lynn and Inner Dowsing. "The area of London Array OWF is
characterized by an alternating pattern of deep channels (Black Deep, Knock Deep) and sandbanks (Long Sands,
Kentish Knock). These topographic features significantly contribute to the local scour patterns. Water depths at this
site range from 0 to 30 m, with Long Sands known for its significant variations in bed elevation but general stability
of position. Meanwhile, Knock Deep is notable for its eastward shift over time, which has widened the channel and
maintained a constant bed level.







**Figure 11: A) Spatial distribution of relative scour depths ($S/D$) at London Array OWF from 2010-2014. B) Grain-size distribution. C) Relative scour depths vs relative water depths, and Grain size classification. D) Significant wave heights and E) Current velocities. $S/D$ and $D_{50}$ data are used from Melling (2015)**

In Figure 11A, the distribution of scour depths shows that the variation in scour is strongly influenced by the underlying topography, with significantly greater scour depths on the sand banks compared to the channel. Addionally a trend of increasing scour depths is observed from northeast to southwest, which is particularly notable in the channel area. The smallest scour is observed in the northern part of Knock Deep with a ratio of 0.2 $S/D$ and the largest in the southern part of Long Sands with 2.1 $S/D$. The differences in scour depths can be derived directly from the seabed topography, with greatest average scour depths found in the Long Sands with 1.53 $S/D$, followed by Kentish Knock ($S/D = 1.37$), and then Knock Deep ($S/D = 0.77$) with the smallest average. The sediment distribution across this OWF, shown in Figure 11B, ranges from very fine to coarse sands. Coarse sands can be found in Knock Deep, where generally the smallest scour depths are seen (e.g., L11, J10 and J11). Furthermore, the largest scour depths are noticed in the southern part of Long Sands (e.g. A13-A15, D15-D19, J18 and L18), where the sediment varies from very fine to fine medium sands. There is therefore a reasonable correlation between grain size and scour depth, which is consistent with the previously observed global trend. Additionally, Figure 11C shows a negative correlation between S/D and h/D aligning with the global trend observed in Figure 8, i.e. that shallower relative water depths can be associated with deeper scour, while deeper waters tend to have reduced scour depths. This trend may be explained by the findings of Hjort (1975), who demonstrated that bed shear stress decreases with increasing water depth for the same flow and structure diameter, potentially leading to reduced scour at greater depths. However, as the water depth in the London Array OWF changes simultaneously with the sediments, i.e. coarser grained sediments are present in the deeper water depths of Knock Deep, the cause of the different scour depths cannot be clearly attributed to either the sediments or the water depth. Other hydrodynamic, environmental, and topographic factors also play a critical role in shaping these patterns at this OWF, underscoring the complexity of the influences involved.

Significant wave heights and current velocities, as shown in Figures 11D and 11E, provide important insights into the scour dynamics at the London Array. These figures show that, in addition to relative water depths and sediment grain sizes, wave and current dynamics might be critical factors at this wind farm. The predominant direction of both waves and currents is northeast to southwest, consistent with the estuarine influence of the area, where river discharge also significantly affects hydrodynamic conditions. This influence is particularly evident at the Long Sands and Kentish Knock sandbanks, which are shaped by the combined action of waves and currents (London Array Ltd, 2005).

Figure 11D shows that the highest wave heights are observed coming from the northeast, with values exceeding 3.0 m, and lower wave heights propagating from the southwest. This gradient in wave height suggests a correlation with increased scour depths in regions exposed to higher wave energy, suggesting a strong link between wave dynamics and seabed modification. Similarly, Figure 11E highlights a larger number of strong currents coming from the southeast. These higher velocities correspond to areas with more pronounced scour depths, highlighting the role of strong currents in influencing sediment transport and depositional patterns.

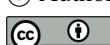



In addition, the local tidal dynamics vary significantly across the wind farm, with the flood tide dominating the southern
banks and the ebb tide more influential on the northern banks (Kenyon and Cooper, 2005). This variation is due to the
sheltering effect of the sandbanks, which are slightly offset from the orientation of the ebb tide, and is particularly
pronounced at Long Sands (London Array Ltd, 2005). The interplay of river discharge, wind stress, tidal surge and
density driven currents follow the pathways created by the existing topography, further complicating the hydrodynamic
environment and its effect on scour at the London Array OWF.
After analyzing the scour depths at 9 wind farms and with different ranges of scour depths, the variation of scour depths
can also be noticed in individual OWFs, as in the case for London Array OWF.

## 4. Implications for scour predictions for OWFs

Overall, this study extends the investigation of scour dynamics to a regional scale by analyzing correlations between
relative scour depth and site conditions across multiple OWFs to identify consistent scour patterns and trends. The
analysis highlights a significant correlation between larger scour depths and finer sediment types (particularly fine and
medium sands), a finding consistent with sediment transport theories that suggest finer noncohesive grains are more
susceptible to mobilization by hydrodynamic forces. This broad correlation, observed across different geographic
locations and environmental conditions, reinforces the universality of sediment size as a fundamental factor in scour
processes, as documented in the extensive work of Vanhellemont et al. (2014) and Rivier et al. (2016). Given the large
underlying database, this study adds weight to the argument for the universal incorporation of detailed sediment
characteristics into scour assessment practices.
However, the analysis also showed that the strong influence of the erosion potential of the sediments in the field alone
cannot describe all observations. The PCA analysis also provided a strong negative correlation observed between
relative scour depth ($S/D$) and relative water depth ($h/D$), particularly in fine and medium sand sediments, suggesting
that the relative water depth ($h/D$) plays a critical role in scour processes, confirming the trends observed by
Whitehouse et al. (2010) and Melling et al. (2015) for field data. The decrease of the scour depth with decreasing water
depth seems unexpected and contradicts common scour prediction approaches like e.g. Breusers et al. (1977), which
however are often derived for flow conditions with shallow water depth. Harris and Whitehouse (2014) argued that in
deeper water, a weaker downflow and hence a weaker horseshoe vortex can be expected, ultimately leading to smaller
scour depth. As the depth increases, the hydrostatic component of the total energy at the front of the pile increases
relative to the kinetic component. Additionally, the reduction in boundary layer thickness, induced by decreased water
depths, has the potential to enhance bed shear stresses, thereby increasing sediment mobility. For certain sediment
groups, the PCA demonstrated a stronger correlation with the pile Reynolds number or the Froude number. This finding
underscores a complex dynamic that is frequently oversimplified in existing models. The results indicate a necessity
to incorporate nonlinear hydrodynamic models into scour prediction frameworks. The results of the PCA reveal the
necessity for a diversified approach to the modeling of scour in complex field conditions, which extends beyond the
scope of traditional uniform applications.
This analysis demonstrates that individual OWFs exhibit unique environmental and sediment conditions, which can
either amplify or moderate broader trends. The London Array OWFs serves as a prime example of the predictive
reliability of observed regional trends, as local data closely mirrors general trends. Conversely, sites such as Robin
Rigg and Lynn and Inner Dowsing exhibit deviations from these trends due to their distinct sediment compositions




and hydrodynamic conditions, underscoring the necessity for site-specific adjustments to scour prediction models.
These findings underscore the intricacy of employing global models on a local scale and underscore the significance
of site-specific data in validating and refining these models to enhance their accuracy and applicability.

**5 Limitations and future research**

Although this study provides a detailed analysis of scour depths at nine OWFs, certain limitations must be addressed
to improve the interpretation of the findings. Although the dataset spans multiple years, it represents snapshots in time
and may not fully capture the dynamic evolution of scour processes under fluctuating metocean conditions (Matutano
et al., 2013; Carpenter et al., 2016). Hindcast data, while valuable for long-term trends, are often based on limited
spatial resolution that may underestimate short-term extreme events such as storm surges or localized current variations
(Whitehouse et al., 2010; Sturt et al., 2009).
Using PCA is effective in identifying dominant linear relationships between scour depths and key variables; however,
it may miss critical nonlinear interactions that drive scour processes (Schendel et al., 2020; Lyu et al., 2021).
Parameters such as the $KC$ number and Shields parameter, which account for sediment motion initiation and
hydrodynamic forces, could not be reliably determined in this study due to data limitations. Given their importance in
understanding sediment transport and scour development (Sheppard et al., 2004; Zhao et al., 2012), future studies
should prioritize the inclusion of these dimensionless parameters to provide a more robust assessment and comparison
of scour processes.
The next step in this research is to develop data-driven models and investigate the broader implications for regional
sediment dynamics. Future studies will focus on OWFs located in fine and medium sands where significant scour
activity is observed. By focusing on these environments, we aim to improve prediction capabilities and better
understand the mechanisms that drive scour, particularly in areas that are susceptible to substantial sediment
mobilization.
Finally, while the present study focused on localized scour processes, the cumulative effects of OWF structures on
regional sediment transport and marine ecosystems remain a significant knowledge gap (Christiansen et al., 2022;
Schultze et al., 2021). Future research must employ interdisciplinary methodologies to rigorously assess the ecological
impacts of sediment mobility and scour on marine habitats. By integrating regional sediment transport models with
comprehensive ecological assessments, we can optimize offshore wind energy development to meet both sustainability
and environmental protection goals, ensuring long-term benefits for infrastructure resilience and marine ecosystem
health.

**6 Conclusion**

Achieving the European Union's (EU) offshore wind energy targets requires development of OWFs in regions with
diverse and often poorly understood meteoceanic and geophysical conditions. However, this demand underscores
critical knowledge gaps regarding the interaction of these installations with the marine environment, particularly with
respect to scour processes and sediment mobilization. A comprehensive understanding of scour dynamics is essential,
not only to ensure structural integrity, but also to assess potential impacts on regional sediment transport and broader
ecosystem functions.





In this study, high-resolution bathymetry data were used to analyze field-measured scour depths of 460 monopiles
across nine British OWFs. The analysis included a PCA in which eight hydrodynamic and geotechnical variables were
considered to identify the dominant driver influencing scour depths variability. This analysis provided a basis for
understanding the primary correlations between scour depths and metocean site conditions, but also highlighted the
complexity of these relationships, requiring further refinement.
The main conclusions can be summarized as follows:

(1) **Universal drivers of scour:** Across all nine OWFs, the PCA identified $D_{50}$, as one of the main drivers
in influencing scour depths variability, together by the relative water depths ($h/D$). The analysis across
all OWFs (Fig. 8) showed that greater scour depths occurred in shallower waters, particularly at location
with sediments composed of (63 to 200 $\mu m$) and medium sand (200 to 630 $\mu m$). This result highlights
the critical role of sediment size in scour formation and confirms that finer sediments are more susceptible
to hydrodynamic forcing.

(2) **Sediment-specific trends:** In order to explore the variability within sediment types, the data set was
clustered according to $D_{50}$, and a PCA was applied to each cluster. For fine sand (63 to 200 $\mu m$) and
medium sand (200 to 630 $\mu m$), relative water depths was found to be the dominant driver of scour depths,
demonstrating the sensitivity of these sediment types to hydrodynamic forcing in shallower relative water
depths. For coarser sediments, such as coarse sands (630 to 2000 $\mu m$) and fine gravels (2000 to 6300
$\mu m$), the correlations were less pronounced, reflecting a greater resistance to scour. This sediment-
specific analysis highlights the importance of considering sediment type when assessing scour
susceptibility and designing OWFs, and how different sediment types can influence sediment transport
patterns.

(3) **Site-specific variability:** Due to local factors such as sediment conditions, hydrodynamic conditions,
and topography, individual OWFs exhibited unique scour depths patterns. For example, London Array
(Fig. 11C) showed trends similar to the global results (Fig. 8), with relative water depths and site
topography as the primary influences on scour, followed by current and wave conditions. In contrast,
OWFs such as Robin Rigg and Lynn and Inner Dowsing showed no discernible trends between scour
depths and the key drivers obtained from the global PCA, highlighting the need for individual analyses
to account for local complexities.

This study also highlights the potential environmental impacts of scour-induced sediment transport. While the primary
focus was on identifying the physical drivers of scour, the findings could provide a first step in assessing potential
impacts of OWF on the marine environment due to a changed regional sediment mobility. The entrainment of eroded
sediment into the water column, with subsequent long-range transport, raises concerns about sediment deposition and
potential impacts on benthic habitats and marine wildlife in far-field regions.
Future research should prioritize the refinement of predictive scour models that incorporate temporal data and
expanded hydrodynamic parameters to improve accuracy in diverse sedimentary environments. In addition, integrated
approaches that combine regional sediment transport modeling with ecological assessments are critical for evaluating



the cumulative impacts of OWF facilities on marine ecosystems. These efforts will facilitate the development of
sustainable OWF designs that minimize environmental disturbance while advancing renewable energy goals.
**Data availability:** The data set used in this study is available in the Marine Data Exchange (MDE)
(https://www.marinedataexchange.co.uk/) and by the Copernicus Marine Service (CMEMS)
(https://marine.copernicus.eu/)

**Authors contribution: K. G.:** Writing – original draft preparation, visualization, formal analysis, conceptualization,
methodology. **C.J.** Writing – review & editing, supervision, conceptualization, project administration. **G.M**. Writing
– review & editing, resources. **A.S.** Writing – review & editing, methodology. **M.W**. Writing – review & editing,
Supervision. **T.S.** Writing – review & editing, Funding acquisition, Supervision.
**Competing interests:** The authors declare that they have no conflict of interest**.**
**Acknowledgements:** This work contributes to the DAM Research Mission sustainMare and the project CoastalFutures
(Project Number: 03F0911G) funded by the German Federal Ministry of Education and Research (BMBF). The
responsibility for the content of this publication lies with the authors.

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
