# Peer review of "Scour variability across offshore wind farms (OWFs): Understanding site-specific scour drivers as a step towards assessing potential impacts on the marine environment"

_Wind Energy Science, 2025_

## Author Comment (AC1)

**Response to reviewers**

Preprint wes-2025-41

Title: Scour variability across offshore wind farms (OWFs): Understanding site-specific scour drivers as a step towards assessing potential impacts on the marine environment

Authors: Karen Garcia, Christian Jordan, Gregor Melling, Alexander Schendel, Mario Welzel, Torsten Schlurmann

**Authors' reply to comments**

We sincerely thank the reviewer for taking the time to evaluate our work and for recognizing its contributions. After carefully reading and discussing the remarks, we have thoroughly revised and improved the manuscript accordingly. Please find our responses (blue) and revised text blocks (*blue, italic*) below the quoted reviewer comments (**black**). Please note that, in the meantime, we have revised the data, added new equations and generated new figures to improve the manuscript based on the reviewer's comments. Moreover, we included a few minor changes that might not refer directly to specific reviewer comments but are meant to enhance the readability and hence understanding of our approach and findings according to a native speaker.

**Response to Referee #1**

The paper investigates the variability of scour depths around monopile foundations in 1. offshore wind farms (OWFs) and identifies the key drivers influencing these scour processes. The study utilizes high-resolution bathymetry data from 460 monopiles across nine British OWFs to analyze spatial and environmental factors affecting scour. It should be acknowledged that the measurement data from the present study are of great engineering value to predict the scour before the planning of OWF. The methodology of interpreting the data is reasonable, and the general conclusion from the interpretation is plausible. The primary problem is that the overall manuscript lacks physical analysis and correlation of the scour depth with the hydrodynamic quantities. This is understandable due to the lack of critical Shield parameters, which are directly linked to the scour process. Furthermore, in shallow water regions, it seems that the wave characteristics associated with the KC number play an important role in determining the scour depth. As pointed out by the authors, this parameter is not analyzed due to the limited measurement. Despite this lack, the physical interpretation of this manuscript can still be enhanced by estimating the values of the Shields and KC based on some assumptions. For example, combined with Re and the roughness associated with the grain size, the shear stress can be estimated based on a log law assumption. Combined with the dispersion relationship, knowing the water depth, the waveinduced velocity can be estimated so that the KC value can be estimated.

**Dear Referee #1,**

1. The authors appreciate the reviewer's most valuable comment and appreciate the recognition of the value of our study in analyzing the variability of scour depths around monopile foundations at offshore wind farms (OWFs). We are pleased that the reviewer acknowledges the engineering value of the high-resolution bathymetry data of 460 monopiles at nine UK OWFs and the methodology used to interpret these data.

We fully concur with the critical observation regarding the need for a more in-depth physical analysis, specifically the correlation of scour depths with key hydrodynamic parameters such as the Shields parameter ( $\theta_{99}$ ) and the Keulegan-Carpenter number ( $KC_{99}$ ). The reviewer correctly identified these parameters as fundamental to the scour process. We acknowledge that their absence from the initially submitted manuscript limits the ability to interpret the findings systematically.

In response, equations 9 and 19 were used and are included in Table 2 on pages 10-11 to calculate the  $KC_{99}$  and  $\theta_{99}$ , respectively:

| Variable                                                       | Equation                                                                                                                                                                                                                                     |               |
|----------------------------------------------------------------|----------------------------------------------------------------------------------------------------------------------------------------------------------------------------------------------------------------------------------------------|---------------|
| Zero crossing period $(T_z)$                                   | $T_z = \frac{T_p}{1.28}$                                                                                                                                                                                                                     | (8a)          |
| Natural scaling period
( T n )            | $T_n = \sqrt{\frac{h}{g}}$                                                                                                                                                                                                                   | (8b)          |
| $(A_t)$                                                        | $A_t = (6500 + \left(0.56 + 15.54 \frac{T_n}{T_z}\right)^6)^{1/6}$                                                                                                                                                                           | (8c)          |
| RMS velocity ( U rms )                       | $U_{rms} = 0.25 \frac{H}{T_n (1 + \left(A_t \left(\frac{T_n}{T_z}\right)^2\right))^3}$                                                                                                                                                       | (8 d ) |
|                                                                |                                                                                                                                                                                                                                              | (8e)          |
| Wave-induced velocity                                          | $U_m = \sqrt{2} U_{rms}$                                                                                                                                                                                                                     |               |
| ( U m )                                      |                                                                                                                                                                                                                                              |               |
| Keulegan-Carpenter
number ( KC )                     | $KC_{99} = \frac{U_m T_p}{D}$                                                                                                                                                                                                                | (9)           |
| Roughness related to
d 50 (ks)                   | $k_{s} = 2.5d_{50}$                                                                                                                                                                                                                          | (10)          |
| Amplitude of wave
orbital motion at the bed
( A ) | $A = \frac{U_m T_p}{2\pi}$                                                                                                                                                                                                                   | (11)          |
| shear velocity $(\boldsymbol{U}_f)$                            | $U_f = \frac{U}{6.0 + 2.5 \ln\left(\frac{h}{k_s}\right)}$                                                                                                                                                                                    | (12)          |
| wave friction factor
( f w )              | $f_w = \begin{cases} 0.32 \left(\frac{A}{k_s}\right)^{-0.8}, & \frac{A}{k_s} < 2.92\\ 0.237 \left(\frac{A}{k_s}\right)^{-0.52}, 2.92 \le \frac{A}{k_s} < 727\\ 0.04 \left(\frac{A}{k_s}\right)^{-0.25}, & \frac{A}{k_s} \ge 727 \end{cases}$ | (13)          |
| angular difference
between the direction of                 | $\alpha = atan2(u_0, v_0) - D_w$                                                                                                                                                                                                             | (14)          |

Table 2. Calculation of the variables included in the analysis

| the wave and the current $(\alpha)$                                                             |                                                                                              |      |
|-------------------------------------------------------------------------------------------------|----------------------------------------------------------------------------------------------|------|
| current induced bed shear stress ( $	au_c$ )                                                    | $	au_c =  ho_w U_f^2$                                                                        | (15) |
| wave induced bed shear stress $(\boldsymbol{\tau}_{\boldsymbol{w}})$                            | $\tau_w = 0.5 \rho_w f_w U_m^2$                                                              | (16) |
| cycle-mean shear stress $(\boldsymbol{\tau}_{m})$ due to a combined wave-current load           | $\tau_m = \tau_c \left[ 1 + 1.2 \left( \frac{\tau_w}{\tau_c + \tau_w} \right)^{3.2} \right]$ | (17) |
| maximum shear stress
value under combined
wave-current load
( tmax ) | $\tau_{max} = \left[ (\tau_m + \tau_w \cos \alpha)^2 + (\tau_w \sin \alpha)^2 \right]^{0.5}$ | (18) |
| Shields parameter ( $\boldsymbol{\theta}$ )                                                     | $\theta_{99} = \frac{\tau_{max}}{(\rho_s - \rho_w)gd_{50}}$                                  | (19) |
| Mobility parameter $\theta_{99}/\theta_{cr}$                                                    | $\theta_{99}/_{\theta_{cr}}$                                                                 | (20) |

Lines 220-235 which follow Table 2, have been modified to describe the assumptions, physical meaning and relevance of these new parameters:

The values assumed for all OWFs sites are:

$$\rho_s = 2650 \ kg/m^3, \ \rho_w = 1027 \ kg/m^3, \ v = 1.3 x 10^{-6} m^2/s, \ g = 9.8 \ m/s^2$$

Where  $\rho_s$  is the sediment density, based on Soulsby (1997).  $\rho_w$  is the water density, v is the kinematic viscosity and g the gravitational acceleration. Equation 4 was calculated based on van Rijn (1984), where  $D_*$  is the non-dimensional grain diameter that is used to calculate the critical Shields parameter ( $\theta_{cr}$ ), which represents the threshold for initiation of motion at the bed, as proposed by Soulsby (1997).

Equation 5 is taken from Soulsby and Whitehouse (1997), where  $s (s = \rho_s / \rho_w)$  represents the specific gravity of sediment grains. The  $d_{50}$  represents the median sediment grain size.

In equation 18, the maximum bed shear stress value ( $\tau_{max}$ ) was calculated following Roulund et al. (2016), which builds upon Soulsby (1997) by combining current- and wave-induced shear stress through a directional correction. Shields parameter ( $\theta_{99}$ ) is derived using equation 19, based on the maximum bed shear stress ( $\tau_{max}$ ) under combined wave and current conditions. The Keulegan–Carpenter number (KC99) is defined in equation 10, where  $T_p$  is the peak wave period and D the monopile diameter.

Equation 20 provides the calculation of the mobility parameter ( $\theta_{99}/\theta_{cr}$ ) to assess sediment mobility, providing a dimensionless indicator of whether the hydrodynamic forcing was sufficient to initiate sediment motion. All relevant equations are summarized in table 2.

Due to the consideration of parameters such as Keulegan-Capenter number, and sediment mobility  $(\theta_{99}/\theta_{cr})$  parameter, together with the concern of reviewer 2, in regards of the choice of a mix of dimensionless and dimensional parameters in the PCA, The figure 4, which represents the PCAhas been updated with the considerations of just dimensionless parameters (Figure 5 and 6 have been also updated, and added in the manuscript), see in page 18:

Figure 4: a) PCA biplot, illustrating the correlation between dimensionless variables and relative scour depths. b) The table details the angles between S/D and the other variables (in degrees), along with the magnitude cosine-based correlation (values from 0 to 1), where values closer to 1 indicate stronger correlation.

Lines 3368-410 (Page 18 -19), which describe the results from the PCA, have been modified:

"As shown in the biplot, PC1 and PC2 account for 74.03% of the variation in the data set. This high percentage indicates that these two components capture most of the significant patterns in the data, allowing for a meaningful interpretation of the relationships among the variables. In the biplot, each vector stands for a variable, with the direction and magnitude of the vector reflecting its contribution to the principal components. The variables that contribute the most to the variance in PC1 are the mobility parameter, the Froude number, and Keulegan Carpenter number, with shares of 0.4898, 0.4419, and 0.4114, respectively. In contrast, the variance in PC2 is primarily explained by the pile Reynolds number (, the relative grain size and the Froude number, with shares of 0.628, -0.489, and 0.3168, respectively. This significant contribution of the mobility parameter, the Froude number, and the Keulegan Carpenter number, the Froude number, and the Keulegan Carpenter number to PC1

suggests that variations in these hydrodynamic parameters are critical in shaping the principal dynamics of the dataset.

The table (Fig. 4b) next to the biplot provides further insight by showing the angular distances between the *S/D* vector and each of the other variables, as well as their respective correlation coefficients. One of the key observations is that the relative scour depth has the strongest negative correlation of 0.96 with the relative water depth, which underscores the critical role of water depth in governing scour intensity. Shallower relative depths concentrate flow energy at the bed, intensifying near-bed velocities and shear stresses that promote deeper scour holes (Smith & McLean, 1977; Whitehouse, 2010). The next strongest correlation is with the relative grain size with a correlation factor of 0.81. This suggests that as the relative grain size increases, relative scour depth also tends to increase. This trend is in line with the functional dependence of relative scour depth on relative grain size as observed by Sheppard et al. (1995, 1999). This positive trend may be due to increased turbulence caused by larger bed roughness elements or the initiation of larger-scale scour processes around coarser particles under certain flow conditions (Whitehouse, 2010).

Furthermore, a significant positive correlation was found with the Keulegan-Carpenter number with a correlation factor of 0.81, indicating the importance of oscillatory flow conditions in scour development. Higher Keulegan Carpenter numbers directly lead to higher relative scour depths (Sumer and Fredsoe, 2002). This is driven by the onset of the horseshoe vortex and lee-wake eddy shedding (Sumer et al., 1992b; Zanke et al., 2011), with increased permanence of the horseshoe vortex and amplification of bed shear stresses at higher KC values (Sumer et al., 1997). In addition, the mobility parameter exhibits a strong positive correlation (0.71) with the relative scour depth. The mobility parameter quantifies the instantaneous capacity of the flow to exceed the entrainment threshold, driving rapid sediment entrainment when significantly above unity (Soulsby, 1997; van Rijn, 1993). Variables such as the pile Reynolds number, the flow intensity , and the Froude number, although less correlated with relative scour depths, contribute more to the total variance. This suggests that these flow-related variables influence relative scour depths through more complex or non-linear interactions with other hydrodynamic conditions and sediment characteristics.

Since seabed sediment characteristics play a significant role in local scour (Qi et al., 2016), the PCA was applied again to the same dataset but pre-clustered into different soil classes (Annad et al. 2021). By reducing the uncertainties related to the grain size ( $d_{50}$ ), this analysis should provide a better estimation of the local scour. This classification also facilitates the identification of parameters that are more influential in estimating scour for specific soil classes rather than uniformly across different types. After the clustering, six soil classes were obtained: cohesive sediment ( $d_{50} \le 63 \ \mu m$ ) with 5 data points, fine sand ( $63 \le d_{50}

---

## Author Response (AR1)

**Authors' reply to comments – Referee 1**

Preprint wes-2025-41

Title: Scour variability across offshore wind farms (OWFs): Understanding site-specific scour drivers as a step towards assessing potential impacts on the marine environment

Authors: Karen Garcia, Christian Jordan, Gregor Melling, Alexander Schendel, Mario Welzel, Torsten Schlurmann

We sincerely thank the reviewer for taking the time to evaluate our work and for recognizing its contributions. After carefully reading and discussing the remarks, we have thoroughly revised and improved the manuscript accordingly. Please find our responses (blue) and revised text blocks (*blue*, *italic*) below the quoted reviewer comments (**black**). Please note that, in the meantime, we have revised the data, added new equations and generated new figures to improve the manuscript based on the reviewer's comments. Moreover, we included a few minor changes that might not refer directly to specific reviewer comments but are meant to enhance the readability and hence understanding of our approach and findings according to a native speaker.

**Response to Referee # 1**

The paper investigates the variability of scour depths around monopile foundations in offshore wind farms (OWFs) and identifies the key drivers influencing these scour processes. The study utilizes high-resolution bathymetry data from 460 monopiles across nine British OWFs to analyze spatial and environmental factors affecting scour. It should be acknowledged that the measurement data from the present study are of great engineering value to predict the scour before the planning of OWF. The methodology of interpreting the data is reasonable, and the general conclusion from the interpretation is plausible. The primary problem is that the overall manuscript lacks physical analysis and correlation of the scour depth with the hydrodynamic quantities. This is understandable due to the lack of critical Shield parameters, which are directly linked to the scour process. Furthermore, in shallow water regions, it seems that the wave characteristics associated with the KC number play an important role in determining the scour depth. As pointed out by the authors, this parameter is not analyzed due to the limited measurement. Despite this lack, the physical interpretation of this manuscript can still be enhanced by estimating the values of the Shields and KC based on some assumptions. For example, combined with Re and the roughness associated with the grain size, the shear stress can be estimated based on a log law assumption. Combined with the dispersion relationship, knowing the water depth, the wave-induced velocity can be estimated so that the KC value can be estimated.

**Dear Referee #1,**

1. The authors appreciate the reviewer's most valuable comment and appreciate the recognition of the value of our study in analyzing the variability of scour depths around monopile foundations at offshore wind farms (OWFs). We are pleased that the reviewer acknowledges the engineering value of the high-resolution bathymetry data of 460 monopiles at nine UK OWFs and the methodology used to interpret these data.

We fully concur with the critical observation regarding the need for a more in-depth physical analysis, specifically the correlation of scour depths with key hydrodynamic parameters such as the Shields parameter  $(\theta_{99})$  and the Keulegan-Carpenter number  $(KC_{99})$ . The reviewer

correctly identified these parameters as fundamental to the scour process. We acknowledge that their absence from the initially submitted manuscript limits the ability to interpret the findings systematically.

In response, equations 9 and 19 were used and are included in Table 2 on pages 10-11 to calculate the  $KC_{99}$  and  $\theta_{99}$ , respectively:

*Table 2.* Calculation of the variables included in the analysis

| Variable                                                                 | Equation                                                                                                                                                                                                                                                     |      |
|--------------------------------------------------------------------------|--------------------------------------------------------------------------------------------------------------------------------------------------------------------------------------------------------------------------------------------------------------|------|
| Zero crossing period $(T_z)$                                             | $T_z = \frac{T_p}{1.28}$                                                                                                                                                                                                                                     | (8a) |
| Natural scaling period $(T_n)$                                           | $T_n = \sqrt{rac{h}{g}}$                                                                                                                                                                                                                                    | (8b) |
| $(A_t)$                                                                  | $A_t = (6500 + \left(0.56 + 15.54 \frac{T_n}{T_z}\right)^6)^{1/6}$                                                                                                                                                                                           | (8c) |
| RMS velocity ( $U_{rms}$ )                                               | $U_{rms} = 0.25 \frac{H}{T_n (1 + \left(A_t \left(\frac{T_n}{T_z}\right)^2\right))^3}$                                                                                                                                                                       | (8d) |
|                                                                          | ,                                                                                                                                                                                                                                                            | (8e) |
| Wave-induced velocity                                                    | $U_m = \sqrt{2} \ U_{rms}$                                                                                                                                                                                                                                   |      |
| $(\boldsymbol{U_m})$                                                     |                                                                                                                                                                                                                                                              |      |
| Keulegan-Carpenter
number (KC)                                        | $KC_{99} = \frac{U_m T_p}{D}$                                                                                                                                                                                                                                | (9)  |
| Roughness related to $d_{50}$ (ks)                                       | $k_{\scriptscriptstyle S}=2.5d_{50}$                                                                                                                                                                                                                         | (10) |
| Amplitude of wave orbital motion at the bed (A)                          | $A = \frac{U_m T_p}{2\pi}$                                                                                                                                                                                                                                   | (11) |
| shear velocity $(U_f)$                                                   | $U_f = \frac{U}{6.0 + 2.5 \ln\left(\frac{h}{k_s}\right)}$                                                                                                                                                                                                    | (12) |
| wave friction factor (f w )                                   | $f_{w} = \begin{cases} 0.32 \left(\frac{A}{k_{s}}\right)^{-0.8}, & \frac{A}{k_{s}}

| Variables                 | θ to S/D | Cosine-based     |
|---------------------------|----------|------------------|
|                           |          | Correlation with |
|                           |          | S/D              |
| h/D                       | 165.59   | 0.96             |
| $D/d_{50}$                | 35.52    | 0.81             |
| KC 99          | 35.91    | 0.81             |
| $\theta_{99}/\theta_{cr}$ | 43.60    | 0.72             |
| Re 99          | 131.21   | 0.65             |
| $(U/U_{cr})_{99}$         | 83.84    | 0.11             |
| Fr 99   | 84.17    | 0.10             |

Figure 4: a) PCA biplot, illustrating the correlation between variables and relative scour depth . b) The table detailing the angles between the relative scour depth and the other variables (in degrees), along with the magnitude cosine-based correlation (values from 0 to 1), where values closer to 1 indicates stronger correlation.

Lines 383-429 (Page 17 - 18), which describe the results from the PCA, have been modified:

"As shown in the biplot, PC1 and PC2 account for 74.03% of the variation in the data set. This high percentage indicates that these two components capture most of the significant patterns in the data, allowing for a meaningful interpretation of the relationships among the variables. In the biplot, each vector stands for a variable, with the direction and magnitude of the vector reflecting its contribution to the principal components. The variables that contribute the most to the variance in PC1 are the mobility parameter, the Froude number, and Keulegan Carpenter number, with shares of 0.4898, 0.4419, and 0.4114, respectively. In contrast, the variance in PC2 is primarily explained by the pile Reynolds number, the relative grain size and the Froude number, with shares of 0.628, -0.489, and 0.3168, respectively. This significant contribution of the mobility parameter, the Froude number, and the Keulegan Carpenter number to PCI suggests that variations in these hydrodynamic parameters are critical in shaping the principal dynamics of the dataset. The table (Fig. 4b) next to the biplot provides further insight by showing the angular distances between the S/D vector and each of the other variables, as well as their respective correlation coefficients. One of the key observations is that relative scour depth has the strongest negative correlation of 0.96 with the relative water depth, which underscores the critical role of water depth in governing scour intensity. Shallower relative depths concentrate flow energy at the bed, intensifying near-bed velocities and shear stresses that promote deeper scour holes (Smith & McLean, 1977; Whitehouse, 2010). The next strongest correlation is with the relative grain size with a correlation factor of 0.81. This suggests that as the relative grain size increases, relative scour depth also tends to increase. This trend is in line with the functional dependence of relative scour depth on relative grain size as observed by Sheppard et al. (1995, 1999). This positive trend may be due to increased turbulence caused by larger bed roughness elements or the initiation of larger-scale scour processes around coarser particles under certain flow conditions (Whitehouse, 2010).

Furthermore, a significant positive correlation was found with the Keulegan-Carpenter number with a correlation factor of 0.81, indicating the importance of oscillatory flow conditions in scour development. Higher Keulegan Carpenter number directly leads to higher relative scour depth (Sumer and Fredsoe, 2002). This is driven by the onset of the horseshoe vortex and lee-wake eddy shedding (Sumer et al., 1992b; Zanke et al., 2011), with increased permanence of the horseshoe vortex and amplification of bed shear stresses at higher KC values (Sumer et al., 1997). In addition, the mobility parameter exhibits a strong positive correlation (0.71) with the relative scour depth. The mobility parameter quantifies the instantaneous capacity of the flow to exceed the entrainment threshold, driving rapid sediment entrainment when significantly above unity (Soulsby, 1997; van Rijn, 1993). Variables such as the pile Reynolds number, the flow intensity, and the Froude number, although less correlated with relative scour depth, contribute more to the total variance. This suggests that these flow-related variables influence relative scour depth through more complex or non-linear interactions with other hydrodynamic conditions and sediment characteristics.

Since seabed sediment characteristics play a significant role to local scour (Qi et al., 2016), the PCA was applied again to the same dataset but pre-clustered into different soil classes (Annad et al. 2021). By reducing the uncertainties related to grain size ( $d_{50}$ ), this analysis should provide a better estimation of the local scour. This classification also facilitates the identification of parameters that are more influential in estimating scour for specific soil classes rather than uniformly across different types. After the clustering, six soil classes were obtained: cohesive sediment ( $d_{50} \le 63 \ \mu m$ ) with 5 data points, fine sand ( $63 \le d_{50}

"Figure 8: Relative scour depth against (a) the relative grain size, and (b) grain size. The red rational polynomial line gives the approximate upper limit of S/D, based on the course of the 99th percentile, for various  $d_{50}$ . Data points for London Array and Thanet OWFs are included from Melling (2015). Figure 8a summarizes the findings from the PCA analysis (Figure 4) by plotting the relationship between the relative scour depth and relative grain size across all the sampled locations. Figure 8b is also shown here to support figure 8a by representing the data in terms of the grain size, allowing the comparison of dimensional and non-dimensional relative grain size. Figure 8a, reveals no clear trend between relative scour depth and relative grain size, indicating that the dimensionless grain size ratio alone does not adequately capture the relationship between sediment properties and scour depth in field data. Sheppard et al. (2004) observed a clear trend of S/D decreasing for  $^{\rm D}/_{d_{50}} > 50$  in laboratory experiments, which is not consistent with our results. However, field data show much weaker dependence due to natural variability in sediment structure and hydrodynamic forcing

On the other hand, Figure 8b illustrates a discernible trend where the largest relative scour depth occurs predominantly in fine to medium sands ( $R^2$ = 0.8407), as indicated by the rational polynomial line which approximates the upper limit of relative scour depth for various grain size. The trend shown in Fig. 8b is well explained. In general, the mobility potential of the sediments decreases with increasing grain size, which leads to lower relative scour depth for coarser sediments. Very fine sediments, on the other hand, are subject to the influence of cohesion forces that reduce their erodibility, which also leads to lower relative scour depth. Therefore, fine and medium sandy sediments have the largest scour potential, which is reflected in the data of Fig. 8b. The different symbols represent the OWF, highlighting the geographic spread and variability within the dataset. However, it is important to note that most of the data points fall within the range of fine to medium sands, potentially skewing the interpretation.

Figure 9: Relative scour depth against the a) Keulegan-Carpenter number and b) the mobility parameter. Red line gives the power fit line based on the 99th percentile of the data of relative scour depth for various  $\mathbf{d_{50}}$ . Data points for London Array and Thanet OWFs are included from Melling (2015).

The third and fourth parameters, that correlate with the relative scour depth, are the Keulegan-Carpenter number and the mobility parameter as identified by the PCA. Figure 9a shows the correlation between the relative scour depth and the Keulegan-Carpenter number, revealing a distinct increase of relative scour depth with increasing Keulegan-Carpenter number up to  $KC_{99} = 0.5$ . Above this value, relative scour depth shows little variation with further increase of the Keulegan-Carpenter number, which reaches a maximum value of 2.5 in this field dataset. Those results are generally consistent with findings from previous studies (e.g., Qu et al., 2024; Sumer & Fredsøe, 2002), which indicate that scour development is strongly dependent on  $KC_{99}$  at lower values, but becomes less sensitive as  $KC_{99}$  increases. However, experimental studies often focus on wave regimes with KC numbers greater than 6, since it has been established that this is the threshold for generating a horseshoe vortex. Despite considering the 99th percentile of KC numbers over the time period in question, the KC numbers are much smaller for the field conditions presented herein. This strengthens the argument for further scour research to focus on boundary conditions with low KC values.

Figure 9b shows the correlation between relative scour depth and mobility parameter, comparing the Shields parameter with its critical threshold for sediment motion, and revealing a distinct increase of relative scour depth with increasing mobility parameter up to approximately  $\theta_{99}/\theta_{cr}=5$ . At higher mobility values (typically above 5–10), the increase in scour depth tends to stabilize. This trend aligns with experimental observations from Sumer et al. (2013), Chiew (1984), and others, which describe similar stabilization of scour depth under fully mobile conditions. Notably, the response also varies with sediment type: coarser sediments exhibit low relative scour depth values even at high mobility ratios, likely due to their higher resistance to entrainment and potential armoring effects. In contrast, finer sediments (e.g.,  $d_{50} < 200 \,\mu\text{m}$ ) show a steeper increase in scour depth, reflecting their greater susceptibility to hydrodynamic conditions.

Overall, Figure 9a and 9b emphasize the nonlinear and sediment-dependent nature of scour formation. The separation of trends by soil class supports the need for sediment-specific scour prediction models, as also suggested in previous studies (e.g., Whitehouse et al., 2011; Sumer & Fredsøe, 2002). The results provide empirical evidence of this dependency using field-scale data, bridging a critical gap between controlled experiments and real-world conditions."

**2. Some other suggestions to improve the manuscript are listed as follows:**

On Page 6, it is stated that the Froude number influences the scour depth. However, Fr is more related to the free-surface waves. How can it be related to the scour depth at the seabed? The analysis of the mechanism, which is stated to be related to the pressure gradients at the pile, is not clear. How can the free-surface waves affect the pressure gradients around the pile? How are the pressure gradients associated with the scour process? The authors should describe these problems in more detail, at least by providing some reference studies.

2. We would like to apologize for the confusion. While free-surface waves will also have a transient effect on the pressure gradient, in our analysis, the Froude number (Fr) is defined in the context of tidal current-induced flows around the monopile and is calculated as follows (see Table 2, equation 2):

$$Fr = \frac{U}{\sqrt{gh}}$$

Where *U* is the near-bed current velocity and *h* is the local water depth.

Large Fr induces intensified inertial forces that produce a strong adverse pressure gradient at the upstream site of the pile. This results in early boundary layer separation from the seabed. The separated shear layer rolls into a horseshoe vortex, and the vortex' core strength increases with Fr. This amplifies bed shear stress around the scour perimeter and accelerates sediment erosion (Hu, 2021).

Laboratory and numerical studies under transitional Reynolds regimes affirm this mechanism. Corvaro et al. (2015) found that increasing Fr results in larger vortex diameters and higher (bed shear stress  $-\tau_b$ ) concentrations, leading to deeper equilibrium scour depths.

In response, lines 201 - 211 on page 6-7 were modified and references as well as the definition of the  $Fr_{99}$  were added:

"The Froude number  $(Fr_{99})$  and pile Reynolds number  $(Re_{99})$  are used to characterize the flow conditions around the pile and their calculations are shown in table 2, Equations 2 and 3. The Froude number indicates whether the flow is dominated by gravitational or inertial forces. With increasing Froude number, stronger inertial forces produce more pronounced pressure gradients at the upstream face of the monopile. Promoting early boundary layer separation and enhances the strength of the horseshoe vortex system near the seabed, which increases local bed shear stress and accelerates sediment erosion. As shown by Hu (2021), these dynamics are key in amplifying scour. Similarly, Corvaro et al. (2015) found that higher Froude numbers lead to larger vortex structures and increased bed shear stress, resulting in deeper equilibrium scour depth. On the other hand, the Reynolds numbers provides information on whether the flow is laminar or turbulent, and determines the characteristics of the vortex system around the pile."

- 3. As an important component of the analysis, the process of performing the PCA should be elaborated in more detail. For example, how to arrange the current measurement data into matrices and how to compute the correlation angle should be introduced.
- 3. We thank the reviewer for this valuable comment. We have provided a better explanation in the methodology section 2.3, where it is explained how the current measurement data was arranged into matrices and, how other studies used the PCA for scour data. See lines 310 321 on page 14:

"

In this study, the PCA was applied to a dataset of 692 OWES, including 460 from our analysis and an additional 232 OWES from London Array and Thanet OWF, based on Melling's (2015) data. The PCA was then performed using eight independent variables that contributed to the principal components. Those dimensionless variables were the relative water depth (h/D), Keulegan-Carpenter number  $(KC_{99})$ , mobility parameter  $(\theta_{99}/\theta_{cr})$ , Reynolds number  $(Re_{99})$ , Froude number  $(Fr_{99})$ , relative sediment size  $(D/d_{50})$ , flow intensity  $(U/U_{cr})_{99}$ , and the relative scour depth (S/D). Following this, the data was organized into a matrix, with each row representing a specific OWES and each column representing a selected dimensionless variable. All the variables were extracted as representative values specific to the OWES, with the focus on the 99th percentile to capture extreme hydrodynamic conditions. Scour processes are more likely to occur in these extreme conditions because maximum scour depth usually develops during storm-induced events, rather than under mean or median values. Subsequently, the variables were standarized to ensure the comparability of the results.

In some studies, the PCA is used for reducing the number of dimensions (Harasti, 2022), or to help develop predictive models grouped by soil classes (Annad, 2023). However, the aim of this study was to keep all the principal components. This approach enabled the full exploration of the interdependence between physical drivers and scour response across sites. To interpret the relationships among the variables, a principal component analysis biplot was generated (Gabriel et al., 1971). In the biplot, variables are represented as vectors, and the angle between vectors indicates the degree of correlation. The strength of the correlation was quantified using the cosine of the angle (Jolliffe & Cadima, 2016), enabling us to assess the strength of association between each variable and scour variability across different OWFs sites. Similar to previous studies that applied PCA for parameter selection in bridge pier or scour formula development (Harasti, 2022; Annad, 2023), this multivariate analysis provides a clearer understanding of which parameters dominate the scour process under real offshore conditions"

- 4. On Page 17, the discussion on the influence of the water depth on the scour depth is not sufficient, especially regarding the unexpected decrease in the scour depth with the increasing water depth. The authors stated that it is expected that a large water depth led to a large boundary layer thickness. However, the boundary layer thickness is not necessarily related to the water depth and more associated with Re and seabed roughness. Furthermore, the discussion on the pressure fields is rather superficial. Why does the increasing water depth result in more uniform pressure fields? Why does thinner boundary layer lead to a large shear stress?
- 4. We thank the reviewer for this valuable comment. We have provided a response where, the unexpected decrease of the relative scour depth (S/D) with the increasing relative water depth (S/D) is explained, additional references were added. In response lines 476 498 (page 20 21) were modified:

"In contrast, relative water depth has a strong negative correlation with relative scour depth in fine sand (Figure 5b) and medium sand (Figure 5c). This indicates that as relative water depth increases, relative scour depth tends to decrease in these finer soil classes. From a physical view, Melling (2015) found out that in similar substrates, relative scour depth agree well between different geographic locations and showed that OWES located in sandy sediments exhibit a strong influence of relative water depth on scour, suggesting geotechnical factors are less influential in coarser sediments. Although the observation that relative scour depth decreases as relative water depth increases might initially seem counterintuitive. This behavior is best explained through the transition between shallow-water and deep-water flow regimes. As flow approaches a pile, stagnation pressure develops on its upstream face, causing the flow to separate into an up-flow and a down-flow component. The down-flow is directed toward the bed and promotes the formation of a horseshoe vortex. Flow separation occurs at the stagnation point, defined as the location of maximum energy from the approaching flow at the pile face. The energy of the approach flow consists of hydrostatic and kinetic components, whose vertical distribution is governed by the boundary layer. In shallow water, the kinetic component dominates over hydrostatic pressure, resulting in a stagnation point located higher up the pile, near the water surface. This enhances down-flow and vortex activity, intensifying scour processes (Melville, 2008). Additionally, shallower water often features thinner boundary layers with higher velocity gradients near the seabed, potentially leading to greater bed shear stresses and increased sediment mobility. In contrast, in deeper water, hydrostatic pressure becomes more influential, leading to a more uniform pressure field across the pile face and shifting the stagnation point closer to the bed. This results in weaker down-flow and reduced vortex strength, thereby diminishing the scour depth (FHWA, 2012; Harris & Whitehouse, 2014). Furthermore, Link and Zanke (2004) observed that maximum relative scour depth tends to develop more slowly and reach lower values in deeper water depth, even under constant average flow velocity, due to reduced shear velocity over the undisturbed bed. This highlights that the relationship between relative water depth and scour is not necessarily linear."

- 5. It is difficult to understand the influence of Fr on scour depth. Especially, as shown in Figure 6 (e), it seems that there are only two exceptional points, and the correlation is not strong.
- 5. We acknowledge the reviewer's concern about the apparently weak correlation between the Froude number  $(Fr_{99})$  and scour depth, as shown in Figure 6(e). The limited number of data points exhibiting extreme  $Fr_{99}$  values in our extensive dataset makes it difficult to discern a strong statistical relationship through PCA.

In this regard, lines 582–586 (page 24), where the correlation between Fr and scour depth in fine gravel is described, were modified:

"For fine gravel (Figure 6e), the PCA suggests a correlation between relative scour depth and the Froude number, but this is difficult to confirm visually due to the small sample size and narrow Froude number range. Since relative scour depth is comparatively small in this class, relationships are less clear, and parameters like Froude number come to the foreground that were not as prominent in finer

sediments. A broader distribution of Froude number values would be necessary to confirm this more conclusively"

- 6. On Page 30 from Line 592 to 597, it is stated that the wave dynamics play an important role in determining the scour depth. The authors should provide a more detailed discussion on this subject by estimating the KC number based on the wave height and water depth.
- 6. We acknowledge the request of the reviewer for a more detailed discussion of the significant role of wave dynamics on scour depth. As highlighted on Page 30, there is a correlation between higher wave heights from the northeast and increased scour depths. This observation underscores the strong influence of wave energy on seabed modification.

To provide a more quantitative analysis, we incorporated the into our revised analysis (see Table 2, eq. 9). Additionally, lines 882 – 884 on page 38 were modified:

"Figure 12D shows that the highest wave heights are observed coming from the northeast, with values exceeding 3.0 m, and lower wave heights propagating from the southwest. This gradient in wave height suggests a correlation with increased relative scour depth in regions exposed to higher wave energy, suggesting a strong link between wave dynamics and seabed modification. However, estimated KC99 numbers remained relatively low across most sites, indicating limited wave-induced orbital motion near the seabed. This suggests that wave action plays a secondary role in scour development compared to currents. Similarly, Figure 12E highlights a larger number of strong currents coming from the southeast. These higher velocities correspond to areas with more pronounced relative scour depth, highlighting the role of strong currents in influencing sediment transport and depositional patterns."

**References:**

Annad, M., Zourgui, N. H., Lefkir, A., Kibboua, A., and Annad, O.: Scour-dependent seismic fragility curves considering soil-structure interaction and fuzzy damage clustering: A case study of an Algerian RC Bridge with shallow foundations, Ocean Eng., 275, 114157, 2023.

Chiew, Y. M.: Local scour at bridge piers, Publication of: Auckland University, New Zealand, 1984.

Corvaro, S., Mancinelli, A., and Brocchini, M.: Flow dynamics of waves propagating over different permeable beds, Coast. Eng. Proc., 35, 35–35, 2016.

Federal Highway Administration (FHWA): Evaluating scour at bridges (HEC-18, Publication No. FHWA-HIF-12-003), U.S. Department of Transportation, 2012.

Gabriel, K. R.: The biplot graphic display of matrices with application to principal component analysis, Biometrika, 58(3), 453–467, 1971.

Hu, R., Wang, X., Liu, H., and Lu, Y.: Experimental study of local scour around tripod foundation in combined collinear waves—current conditions, J. Mar. Sci. Eng., 9(12), 1373, 2021.

Jolliffe, I. T., and Cadima, J.: Principal component analysis: a review and recent developments, Philos. Trans. R. Soc. A Math. Phys. Eng. Sci., 374(2065), 20150202, 2016.

Link, O., and Zanke, U.: On the time-dependent scour-hole volume evolution at a circular pier in uniform coarse sand, in: Proc. 2nd International Conference on Scour and Erosion (ICSE), 2004.

Matthieu, J., and Raaijmakers, T.: Interaction between offshore pipelines and migrating sand waves, 2012.

Qu, L., An, H., Draper, S., Watson, P., Zhao, M., Harris, J., Whitehouse, R., and Zhang, D.: A review of scour impacting monopiles for offshore wind, Ocean Eng., 301, 117385, 2024.

Roulund, A., Sutherland, J., Todd, D., and Sterner, J.: Parametric equations for Shields parameter and wave orbital velocity in combined current and irregular waves, 2016.

Sheppard, D. M., Zhao, G., and Ontowirjo, B.: Local scour near single piles in steady currents, in: Water Resources Engineering, 1809–1813, ASCE, 1995.

Sheppard, D. M., Ontowirjo, B., and Zhao, G.: Conditions of maximum local scour, in: Proc. Stream Stability and Scour at Highway Bridges, Resources Engineering Conference, 1991–1998, edited by Richardson, E. V. and Lagasse, P. F., ASCE, Reston, VA, 1999.

Smith, J. D., and McLean, S. R.: Spatially averaged flow over a wavy surface, J. Geophys. Res., 82(12), 1735–1746, 1977.

Soulsby, R. L., and Whitehouse, R. J. S.: Threshold of sediment motion in coastal environments, in: Proc. 13th Australasian Coastal and Ocean Engineering Conference and 6th Australasian Port and Harbour Conference (Pacific Coasts and Ports' 97), Christchurch, NZ, Vol. 1, 145–150, Centre for Advanced Engineering, University of Canterbury, 1997.

Sumer, B. M., Petersen, T. U., Locatelli, L., Fredsøe, J., Musumeci, R. E., and Foti, E.: Backfilling of a scour hole around a pile in waves and current, J. Waterw. Port Coast. Ocean Eng., 139(1), 9–23, 2013.

Whitehouse, R. J. S., Harris, J. M., Sutherland, J., and Rees, J.: The nature of scour development and scour protection at onshore wind farm foundations, Mar. Pollut. Bull., 62(1), 73–88, 2011.

Whitehouse, R. J. S., and Harris, J.: Scour prediction offshore and soil erosion testing, 2014.

Zanke, U. C. E., Hsu, T.-W., Roland, A., Link, O., and Diab, R.: Equilibrium scour depths around piles in noncohesive sediments under currents and waves, Coast. Eng., 58(10), 986–991, 2011.

**Authors' reply to comments – Referee 2**

Preprint wes-2025-41

Title: Scour variability across offshore wind farms (OWFs): Understanding site-specific scour drivers as a step towards assessing potential impacts on the marine environment

Authors: Karen Garcia, Christian Jordan, Gregor Melling, Alexander Schendel, Mario Welzel, Torsten Schlurmann

We sincerely thank the reviewer for taking the time to evaluate our work and for recognizing its contributions. After carefully reading and discussing the remarks, we have thoroughly revised the manuscript and made substantial changes to it. Please find our responses (blue) and revised text blocks (*blue*, *italic*) below the quoted reviewer comments (**black**). Please note that, in the meantime, we have revised the data, added new equations and generated new figures to improve the manuscript based on the reviewer's comments. Moreover, we included a few minor changes that might not refer directly to specific reviewer comments but are meant to enhance the readability and hence understanding of our approach and findings according to a native speaker.

**Referee # 2**

Dear Prof. Bachynski-Polić,

1. It has been a pleasure to review this manuscript. The authors present a comprehensive analysis of scour around OWFs. From a large data set of scour and environmental parameters at OWFs in British waters, they identify the main drivers of scour phenomena around the structures. They do so at three levels: (i) considering the overall dataset, (ii) chopping it up into median grain size clusters, (iii) zooming in on specific OWFs to see whether the overall correlations hold locally. The work is novel, interesting (a bit lengthy I must say) and in my opinion highly suitable for WES-audience. Also, the figures are helpful.

My specific concerns are on (i) the choice of a mix of dimensionless and dimensional parameters as the set of explaining parameters, (ii) the use of wave height instead of near-bed wave-induced orbital velocities, (iii) the background and limitations of the flow parameter (Uc,99) and (iv) the implication of the word "understanding" in the title. Please see my explanation below.

Please also see my other points further below. I think the manuscript benefit from a more precise and consistent presentation of the content, perhaps some restructuring, and even shortening. Particularly, using the structure of the conclusions (which I think is clear) to present the goals in the Introduction would be helpful. Overall, based on my review, I recommend "minor revision". Depending on your editorial decision, I would be happy to receive a revised version of the manuscript.

- 1. We would like to thank the reviewer for the valuable and insightful comments. After reading the reviewer's comments, the manuscript was revised to provide more precision and clarity.
- (i) The choice of a mix of dimensionless and dimensional parameters as the set of explaining parameters: The reviewer has indeed highlighted an important issue regarding the methodology. We agree that using a mix of dimensional and non-dimensional parameters can lead to ambiguous interpretations and limit the generalizability of the findings.

In response, we have made a new Principal Component Analysis (PCA) analysis that only considers dimensionless parameters. Dimensional parameters such as wave height  $(H_s)$  and current velocity (U) were removed from the direct PCA input. Additionally, median grain size  $(d_{50})$  was transformed into a dimensionless relative roughness parameter  $(D/d_{50})$ , where D is the monopile diameter. Additionally, we have included the mobility parameter MOB  $(\theta_{99}/\theta_{cr})$  and the Keulegan-Carpenter number  $(KC_{99})$  into our PCA. Accordingly, Figure 4, in page 17 has been updated:

Figure 4: a) PCA biplot, illustrating the correlation between variables and relative scour depth. b) The table detailing the angles between the relative scour depth and the other variables (in degrees), along with the magnitude cosine-based correlation (values from 0 to 1), where values closer to 1 indicates stronger correlation.

Lines 383-429 (Page 17 -18), which describe the results from the PCA, have been modified:

"As shown in the biplot, PC1 and PC2 account for 74.03% of the variation in the data set. This high percentage indicates that these two components capture most of the significant patterns in the data, allowing for a meaningful interpretation of the relationships among the variables. In the biplot, each vector stands for a variable, with the direction and magnitude of the vector reflecting its contribution to the principal components. The variables that contribute the most to the variance in PC1 are the mobility parameter, the Froude number, and Keulegan Carpenter number, with shares of 0.4898, 0.4419, and 0.4114, respectively. In contrast, the variance in PC2 is primarily explained by the pile Reynolds number, the relative grain size and the Froude number, with shares of 0.628, -0.489, and 0.3168, respectively. This significant contribution of the mobility parameter, the Froude number, and the Keulegan Carpenter number to PC1 suggests that variations in these hydrodynamic parameters are critical in shaping the principal dynamics of the dataset. The table (Fig. 4b) next to the biplot provides

further insight by showing the angular distances between the S/D vector and each of the other variables, as well as their respective correlation coefficients. One of the key observations is that relative scour depth has the strongest negative correlation of 0.96 with the relative water depth, which underscores the critical role of water depth in governing scour intensity. Shallower relative depths concentrate flow energy at the bed, intensifying near-bed velocities and shear stresses that promote deeper scour holes (Smith & McLean, 1977; Whitehouse, 2010). The next strongest correlation is with the relative grain size with a correlation factor of 0.81. This suggests that as the relative grain size increases, relative scour depth also tends to increase. This trend is in line with the functional dependence of relative scour depth on relative grain size as observed by Sheppard et al. (1995, 1999). This positive trend may be due to increased turbulence caused by larger bed roughness elements or the initiation of larger-scale scour processes around coarser particles under certain flow conditions (Whitehouse, 2010).

Furthermore, a significant positive correlation was found with the Keulegan-Carpenter number with a correlation factor of 0.81, indicating the importance of oscillatory flow conditions in scour development. Higher Keulegan Carpenter number directly leads to higher relative scour depth (Sumer and Fredsoe, 2002). This is driven by the onset of the horseshoe vortex and lee-wake eddy shedding (Sumer et al., 1992b; Zanke et al., 2011), with increased permanence of the horseshoe vortex and amplification of bed shear stresses at higher KC values (Sumer et al., 1997). In addition, the mobility parameter exhibits a strong positive correlation (0.71) with the relative scour depth. The mobility parameter quantifies the instantaneous capacity of the flow to exceed the entrainment threshold, driving rapid sediment entrainment when significantly above unity (Soulsby, 1997; van Rijn, 1993). Variables such as the pile Reynolds number, the flow intensity, and the Froude number, although less correlated with relative scour depth, contribute more to the total variance. This suggests that these flow-related variables influence relative scour depth through more complex or non-linear interactions with other hydrodynamic conditions and sediment characteristics.

Since seabed sediment characteristics play a significant role to local scour (Qi et al., 2016), the PCA was applied again to the same dataset but pre-clustered into different soil classes (Annad et al. 2021). By reducing the uncertainties related to grain size ( $d_{50}$ ), this analysis should provide a better estimation of the local scour. This classification also facilitates the identification of parameters that are more influential in estimating scour for specific soil classes rather than uniformly across different types. After the clustering, six soil classes were obtained: cohesive sediment ( $d_{50} \le 63 \ \mu m$ ) with 5 data points, fine sand ( $63 \le d_{50}

Figure 5: PCA correlation by clustered soil classes based on the grain size  $(d_{50})$ , remaining parameters that are shown in the biplots are explain in data description (section 2.2). a) Cohesive sediment  $(d_{50} \le 63 \ \mu m)$ . b) Fine sand  $(63 \le d_{50}

Figure 6: Pearson correlation of representative variables obtained by PCA analysis with relative scour depth across different soil classes. a) Cohesive sediment ( $d_{50} \le 63 \ \mu m$ ). b) Fine sand ( $63 \le d_{50} < 200 \ \mu m$ ). c) Medium sand ( $200 \le d_{50} < 630 \ \mu m$ ). d) Coarse sand ( $630 \le d_{50} < 2000 \ \mu m$ ). e) Fine gravel ( $2000 \le d_{50} < 6300 \ \mu m$ ). f) Medium gravel ( $d_{50} \ge 6300 \ \mu m$ ). Clustering of the grain size ( $d_{50}$ ) was based on Annad et. al. (2021).

Additionally, lines 551-586 (Page 22 – 24) have been modified based on the new correlation results:

Considering the small number of data points in this sediment cluster, relative scour depth at locations with cohesive sediments (Fig. 6a) show a moderate positive correlation with the mobility parameter. For the fine and medium sand clusters, the PCA revealed a similarly strong dependence of scour depth on relative water depth. Plotting scour depth against relative water depth now shows a clearer trend and hence dependence for the fine sand sites (Fig. 6b) than for the medium sand sites (Fig. 6c). The Pearson coefficients of -0.57 and -0.86 confirm this difference in the dependence of scour depth on relative water depth. The correlations of the fine and medium sand clusters are supported by a larger number of data points, increasing the reliability of the findings.

For the coarse sand (Fig. 6d), the PCA analysis revealed a negative correlation between relative scour depth and flow intensity. This result directly aligns with the established understanding of live-bed scour behavior in coarse-grained sediments. Once flow intensity surpasses the critical threshold  $(U/U_{LL})_{99}$

>1), the sediment mobilizes, establishing live-bed conditions. In such scenarios, the development of large, well-defined scour holes is consistently observed to be suppressed. This suppression occurs because the continuous transport and replenishment of sediment into the scour region actively works against deep erosion. This dynamic equilibrium of the seabed results in shallower, or inherently more unstable, scour holes when compared to clear-water conditions. In clear-water, where sediment remains immobile, scouring is driven purely by flow-induced vortex action around the structure (Sumer & Fredsøe, 2002; Whitehouse et al., 2011). Consequently, the negative correlation observed in this soil class accurately reflects the inherent limitation of scour growth under the highly mobile conditions characteristic of coarse sandy beds.

For fine gravel (Figure 6e), the PCA suggests a correlation between relative scour depth and the Froude number, but this is difficult to confirm visually due to the small sample size and narrow Froude number range. Since relative scour depth is comparatively small in this class, relationships are less clear, and parameters like Froude number come to the foreground that were not as prominent in finer sediments. A broader distribution of Froude number values would be necessary to confirm this more conclusively."

Following the updated PCA, where relative scour depth (S/D) correlates the most with relative the water depth (h/D), the relative grain size  $(D/d_{50})$ , the Keulegan-Carpenter number  $(KC_{99})$ , and the mobility parameter  $(\theta_{99}/\theta_{cr})$ , Figures 7 and 8 have been updated, and in addition a new figure (Fig 9) was added to the manuscript. Figure 7 shows the correlation between S/D and h/D. Figure 8 shows the correlation between S/D and  $D/d_{50}$  (figure 8a) but also the direct correlation between S/D with  $d_{50}$  (figure 8b), to provide a baseline to understand the influence of grain size independent of the pile diameter. Finally, Figure 9 shows the correlation between S/D with  $KC_{99}$  (Figure 9a) and  $\theta_{99}/\theta_{cr}$  (Figure 9b). Accordingly, lines 615-721 have been updated and restructured (page 25-30)

Figure 7: Relationship between relative scour depth and relative water depth. Symbols indicate the various soil classes that were used for clustering. The red rational polynomial line represents a trend based on the course of the 99th percentile. Data points for London Array and Thanet OWFs are included from Melling (2015).

Figure 7 summarizes the findings from the PCA analysis (Figure 4) by plotting the relationship between the relative scour depth and the relative water depth. Relative water depth has shown to be the parameter with the largest correlation influencing relative scour depth. However, it should be noted that relative water depth has a direct effect on other hydrodynamic parameters. For example, not only is the Froude number formed with the water depth, but relative water depth also significantly determine the potential influence of waves on the development of scour, which in this study has also been considered by the Keulegan–Carpenter numbe. Therefore, it remains unclear whether the influence of relative water depth on relative scour depth is a direct causal factor or an indicator of broader changes in hydrodynamic conditions. Nevertheless, Figure 7 illustrates the comprehensive correlation between the relative scour depth and the relative water depth with the differently colored points representing the studied soils classes.

The trend observed in Figures 6b and 6c is reaffirmed in Figure 7. A distinct relationship exists between the relative scour depth and relative water depth in these two sediment types, i.e. both fine sand  $(63 \le 1)$

 $d_{50}

Figure 8: Relative scour depth against (a) the relative grain size, and (b) grain size. The red rational polynomial line gives the approximate upper limit of S/D, based on the course of the 99th percentile, for various  $d_{50}$ . Data points for London Array and Thanet OWFs are included from Melling (2015).

Figure 8a summarizes the findings from the PCA analysis (Figure 4) by plotting the relationship between the relative scour depth and relative grain size across all the sampled locations. Figure 8b is also shown here to support figure 8a by representing the data in terms of the grain size, allowing the comparison of dimensional and non-dimensional relative grain size. Figure 8a, reveals no clear trend between relative scour depth and relative grain size, indicating that the dimensionless grain size ratio alone does not adequately capture the relationship between sediment properties and scour depth in field data. Sheppard et al. (2004) observed a clear trend of S/D decreasing for  $^{\rm D}/_{\rm d_{50}} > 50$  in laboratory experiments, which is not consistent with our results. However, field data show much weaker dependence due to natural variability in sediment structure and hydrodynamic forcing.

On the other hand, Figure 8b illustrates a discernible trend where the largest relative scour depth occur predominantly in fine to medium sands ( $R^2$ = 0.8407), as indicated by the rational polynomial line which approximates the upper limit of relative scour depth for various grain size. The trend shown in Fig. 8b is well explained. In general, the mobility potential of the sediments decreases with increasing grain size, which leads to lower relative scour depth for coarser sediments. Very fine sediments, on the other hand, are subject to the influence of cohesion forces that reduce their erodibility, which also leads to lower relative scour depth. Therefore, fine and medium sandy sediments have the largest scour potential, which is reflected in the data of Fig. 8b. The different symbols represent the OWF, highlighting the geographic spread and variability within the dataset. However, it is important to note that most of the data points fall within the range of fine to medium sands, potentially skewing the interpretation.

Figure 9: a) Relative scour depth against the Keulegan-Carpenter number. b) Relative scour depth against the mobility parameter. Red line gives the power fit line based on the 99th quantile of the data of relative scour depth for various  $d_{50}$ . Data points for London Array and Thanet OWFs are included from Melling (2015).

"The third and fourth parameters, that correlate with the relative scour depth, are the Keulegan-Carpenter number and the mobility parameter as identified by the PCA. Figure 9a shows the correlation between the relative scour depth and the Keulegan-Carpenter number, revealing a distinct increase of relative scour depth with increasing Keulegan-Carpenter number up to  $KC_{99} = 0.5$ . Above this value, relative scour depth shows little variation with further increase of the Keulegan-Carpenter number, which reaches a maximum value of 2.5 in this field dataset. Those results are generally consistent with findings from previous studies (e.g., Qu et al., 2024; Sumer & Fredsøe, 2002), which indicate that scour development is strongly dependent on  $KC_{99}$  at lower values, but becomes less sensitive as  $KC_{99}$  increases. However, experimental studies often focus on wave regimes with KC numbers greater than 6, since it has been established that this is the threshold for generating a horseshoe vortex. Despite

considering the 99th percentile of KC numbers over the time period in question, the KC numbers are much smaller for the field conditions presented herein. This strengthens the argument for further scour research to focus on boundary conditions with low KC values.

Figure 9b shows the correlation between relative scour depth and mobility parameter, comparing the Shields parameter with its critical threshold for sediment motion, and revealing a distinct increase of relative scour depth with increasing mobility parameter up to approximately  $\theta_{99}/\theta_{cr}=5$ . At higher mobility values (typically above 5–10), the increase in scour depth tends to stabilize. This trend aligns with experimental observations from Sumer et al. (2013), Chiew (1984), and others, which describe similar stabilization of scour depth under fully mobile conditions. Notably, the response also varies with sediment type: coarser sediments exhibit low relative scour depth values even at high mobility ratios, likely due to their higher resistance to entrainment and potential armoring effects. In contrast, finer sediments (e.g.,  $d_{50} < 200 \,\mu\text{m}$ ) show a steeper increase in scour depth, reflecting their greater susceptibility to hydrodynamic conditions.

Overall, Figure 9a and 9b emphasize the nonlinear and sediment-dependent nature of scour formation. The separation of trends by soil class supports the need for sediment-specific scour prediction models, as also suggested in previous studies (e.g., Whitehouse et al., 2011; Sumer & Fredsøe, 2002). The results provide empirical evidence of this dependency using field-scale data, bridging a critical gap between controlled experiments and real-world conditions."

**(ii) The use of wave height instead of near-bed wave-induced orbital velocities:**

This observation is in line with a key point raised by Reviewer #1. We acknowledge the critical role of near-bed, wave-induced orbital velocities in scour processes. For this reason, the Keulegan-Carpenter number ( $KC_{99}$ ) has been calculated with the equation that was added in Table 2 on page 10. Furthermore, this parameter has been considered in the new PCA and can be found in the updated PCA in Figure 4 (page 17) as well as in Figure 9a (page 29).

**(iv) The background and limitations of the flow parameter (Uc,99):**

Thanks to the reviewer for clarifying the issue of  $U_c$ . First,  $U_c$  has been changed to U and with our transition to an exclusively dimensionless PCA, dimensional current velocity (U and, thus,  $U_{99}$ ) is no longer a direct input parameter to the PCA. Instead, its influence is incorporated into the calculation of dimensionless hydrodynamic parameters, such as the  $\frac{\theta_{99}}{\theta_{cr}}$ , which accounts for bed shear stress derived from current and wave induced velocities. This approach ensures that the physical effects of the currents are still represented within our dimensionless framework. The background of the obtention of U is answered in the response to comment 4.

**(iv) The implication of the word "understanding" in the title.**

We agree with the reviewers' reservations about the term "understanding" in the title and suggest to substitute it with the term "identifying".

**SPECIFIC CONCERNS**

- 2. Choice of parameters: It strikes me that many but not all (five out of eight) variables are dimensionless. There seems to be some arbitrariness involved, which makes me wonder about the implications of that for the overall results and conclusions. Why did you decide not to scale the other three: wave height, current velocity and sediment size? From physics-based scaling arguments, I would always expect that dimensionless parameters provide the most meaningful way of explaining dependencies. Please clarify.
- 2. Thanks to the reviewer for bringing this point up. Initially we included dimensionless and non-dimensional parameters together to provide a broad and comprehensive view from the engineering and physical perspective. When including dimensionless parameters such as wave height  $(H_s)$ , current velocity (U) and sediment diameter  $(d_{50})$ , it allowed us to reflect the actual range and scale of the environmental conditions observed across the studied OWFs and thus offering a relevance of our analysis for engineering applications. Furthermore, combining dimensional and dimensionless parameters in the initial PCA helped us to identify potential dependencies between related variables, such as U,  $Re_{99}$ , or  $d_{50}$ .

On the other hand, we agree with the reviewer's observation, that bringing dimensionless parameters together offers a more robust physical framework. As mentioned in previous comments, the new analysis has been updated to only include dimensionless parameters. The calculation of the new dimensionless parameters such as the Keulegan-Carpenter number ( $KC_{99}$ ) and the mobility parameter have been added in Table 2 (page 10-11). Furthermore, the new PCA that contains eight dimensionless parameters can be found in Figure 4 (page 16).

Additionally, the reviewer may find an overview of the dimensionless parameters in lines 313-316 (Page 14):

"Those dimensionless variables were the relative water depths (h/D), the Keulegan-Carpenter number  $(KC_{99})$ , the mobility parameter  $(\theta_{99}/\theta_{cr})$ , the pile Reynolds number  $(Re_{99})$ , the Froude number  $(Fr_{99})$ , the relative sediment size  $(D/d_{50})$ , the flow intensity  $(U/U_{Cr})_{99}$  and the relative scour depth (S/D)."

- 3. To me, it would seem logical to convert wave height (along with wave period, which is now overlooked) into near-bed wave-induced orbital velocities (using linear wave theory). This quantity is more directly related to scour. Please comment.
- 3. We value the reviewer's focus on the significance of near-bed wave-induced orbital velocities for comprehending scour. As we outlined in our response to the reviewer's general comment (ii) and the previous specific point regarding parameter choices, we have incorporated this suggestion. Using linear wave theory, we considered wave height and, importantly, wave period as well as water depth, to estimate near-bed velocities. We then used these velocities to calculate

the Keulegan-Carpenter number ( $KC_{99}$ ), a more physically relevant parameter for characterizing wave-induced scour potential. The KC number is now included in our dimensionless PCA. The calculation of the  $KC_{99}$  number can be seen in Table 2, equation 9, and is described in lines 236 – 243 (page 11):

"In equation 18, the maximum bed shear stress value ( $\tau_{max}$ ) was calculated following Roulund et al. (2016), which builds upon Soulsby (1997) by combining current- and wave-induced shear stress through a directional correction. The Shields parameter ( $\theta_{99}$ ) is derived by using equation 19, based on the maximum bed shear stress ( $\tau_{max}$ ) under combined wave and current conditions. The Keulegan–Carpenter number ( $KC_{99}$ ) is defined in equation 10, where  $T_p$  is the peak wave period and D the monopile diameter. Equation 20 provides the calculation of the mobility parameter ( $\theta_{99}/\theta_{cr}$ ) to assess sediment mobility, providing a dimensionless indicator of whether the hydrodynamic forcing was sufficient to initiate sediment motion. All relevant equations are summarized in Table 2."

- 4. What does the current speed Uc really represent? Is it depth-averaged, or near bed? And would this affect the results? And, furthermore, how is the tidal contribution extracted, by filtering out the wave contributions? Also, I do not see the need to add a subscript "c" for currents, as I do not see another type of velocity-related quantity such as wave current (and it also looks a lot like "cr" used for critical velocity).
- 4. The current velocity (U) used in our study was calculated as the magnitude from the depth-averaged eastward  $u_0$  and northward  $v_0$  velocity components obtained from the hindcast data provided by the Copernicus Marine Service (CMEMS). While near-bed velocity would be more relevant for understanding local scour processes, depth-averaged currents offer a consistent representation of large-scale hydrodynamic forcing across all study sites.

Finally, the subscript "c" was removed and the parameter now is denoted simply as U. Changes based on this comment have been applied in lines 193 - 196 (Page 6):

"The 99th percentile of the current velocity magnitude (U) indicates the resultant of eastward ( $u_0$ ) and northward ( $v_0$ ) tidal flow components, which represent the depth-averaged velocity magnitude, whereas  $U_{cr}$  depicts the critical flow velocity for sediment entrainment. Their ratio is the flow intensity  $({}^{U}/{}_{U_{cr}})_{99})$ ,"

- 5. Thirdly, I wonder whether "Understanding" (used in the title) is really the right term to use here, since the PCA-method does not involve a process-oriented analysis or process-based modelling. In my opinion, when adopting PCA as method, this is more about "Identifying" than "Understanding". Please reconsider.
- 5. We agree with the reviewers' reservations about the term "understanding" in the title and suggest to substitute it with the term "identifying".

"Scour variability across offshore wind farms (OWFs): Identifying site-specific scour drivers as a step towards assessing potential impacts on the marine environment"

**OTHER POINTS (PAPER STRUCTURE, GRAMMAR, SPELLING, MATH, FIGURES)**

GENERAL Regarding writing structure, I find it contusing to see different types of paragraph breaks: (i) with vertical white-spacing. (which seems regular); (ii) with a hard return (which I think are unintended), (iii) with or without a horizontal indent (e.g., lines 120, 125). I expect this to be resolved in the final formatting.

We appreciate the reviewer's comment. The inconsistent breaks and indentations were not intentional and likely resulted from merging sections and editing across different versions. The manuscript was carefully revise to ensure consistency in paragraph breaks throughout the manuscript. We also trust the final typesetting process to proper paragraph alignment and consistency in the final layout.

GENERAL At many instances, the authors refer to a parameter (water depth, scour depth) when they actually mean the scaled version of that parameter (h/D and S/D). Please avoid such loose phrasing, since regarding dependencies this cannot be interchanged.

The word "relative" has been added to the water depth and scour depth throughout the whole manuscript to keep consistency.

L19 sure you want to put symbols in abstract? This leaves me guessing. e.g., that D is pile diameter, which I think is undesirable. Please reconsider.

We appreciate the reviewer's comment and understand the concern regarding the use of parameter symbols in the abstract. However, we believe that the referenced parameters are broadly recognized by the intended readership. Including the symbols allows for a more concise and precise presentation of key results and helps maintain a clear connection between the abstract and the main findings of the study. For these reasons, we kindly propose to retain the symbols in the abstract.

L39 Here you introduce OWES, but later you use the term turbine. Are these intended as synonyms? Please clarify/reconsider terminology.

The terminology was standardized and changed from "turbines" to "OWES" in the whole manuscript.

L52 "but": should be "and" (don't see the contradiction here)

L56, the word "but" has been changed to "and"

L58 "the superposition of": I would rather say "the combined effects of" (superposition suggests adding the separate influences)

L62, "the superposition of" has been changed to the combined effect of"

L75 Curious what you mean with sediment mobility. What about the interaction between structures and seabed patterns such as tidal sand waves?

L81 - 84, the text was updated to include a clarification on sediment mobility and a reference to sand waves:

"...and sediment mobility, resulting in changes to suspended sediment concentrations and wave-induced turbidity plumes (Vanhellemont & Ruddick, 2014). This can also lead to dynamic interactions with migrating seabed features, such as sand waves (Matthieu & Raaijmakers, 2012)."

L78 remove one full stop dot

L86, the full stop has been removed.

L96 (and further) "99 quantile": I think what you mean is what I know as the 99th percentile. To my knowledge, quantile is a more general way of chopping up distributions.

Thanks for pointing this out. We agree that percentile is the more suitable term. The term "quantile" has been changed to "percentile" throughout the manuscript

L97 um should be  $\mu$ m and I find the grain size values a bit odd (why so many digits, which suggests an unrealistic precision). And to my knowledge, the large value (19872  $\mu$ m) is coarse gravel.

L108, the um was replaced to  $\mu m$ . In this study and for continuity, the values of the  $d_{50}$  are being used in  $\mu m$ . On the other hand, the  $d_{50}$  clustering was based on Annad et al. (2014), and medium gravel is considered between 6300  $\mu m$  to 20000  $\mu m$ .

L99 Please add a sentence to prepare for and justify the OWF-site specific analyses in Section 3.5.

L111-L115 a sentence was added to justify section 3.5:

"This analysis aims to (1) identify universal drivers of scour across all sites, (2) assess sediment specific trends by grain size  $(d_{50})$  and (3) evaluate site specific variability at the level of three selected OWFs (Robin Rigg, Lynn and Inner Dowsing and London Array). The site specific analysis in Section 3.5 assesses the robustness of the global correlations under local conditions and provides insight into how local conditions influence scour behavior."

L104 "PCA (Principal Component Analysis)": No need to explain this abbreviation twice (already done on line 99).

L120, the repetition of Principal Component Analysis was deleted.

L113 Really necessary to repeat OWFs here? I think it is not needed and also inconsistent.

In L129, the word OWFs was deleted.

L114 (and further) current velocity magnitudes is also known as current speeds

L130 current velocity magnitudes (U99) reflect the fact that the parameter is based on vector components ( $\sqrt{u_0^2 + v_0^2}$ ), thus being derived from eastward and northward tidal velocities, while current speeds are often used in a general context. The distinction is important, as speeds typically refer only to scalar values, whereas velocity magnitude in our study results from combining two directional components. "Current velocity magnitude" have been kept throughout the manuscript.

L117 (and further) EMODET should be EMODNET

L134, the word EMODET has been changed to EMODNET

L122 omit "the quantile of"

L140, the word "the quantile of" was omitted

L127 "Teeside" should be "Teesside"

L145, the word "Teeside" has been changed to "Teesside"

P5, Figure 1: For consistency and clarity, why not add symbol h to both caption and figure denote water depth also here? (similar to how Hs and Uc are presented)

In P5, Figure 1 was updated, "h" reflects water depth and Uc was changed to U.

L175 Use of bot Ucr and Ucrit: please maintain notational consistency

L195 Ucrit was updated to Ucr

L181 Upon introducing Fr and Re, please either immediately define them in the text below or immediately refer to the expressions where they are defined.

L201-202 The definitions for Fr99 and Re99 can be found in Table 2, equations 2 and 3:

"The Froude number  $(Fr_{99})$  and pile Reynolds number  $(Re_{99})$  are used to characterize the flow conditions around the pile and their calculations are shown in Table 2, Equations 2 and 3."

P7, Table 1: Why µm in boldface? Spelling: Gunfleet Sands

P8, Table 1,  $\mu m$  was changed to the normal format. The spelling error was also corrected.

P8, Table 2, Eq.(1): I do not understand why there is 99 on the left-hand side and no 99 around the quantity on the right-hand side. In my opinion, this should be  $U_c,99 = [ sqrt(u0^2+v0^2) ]$  99

We agree with the reviewer's concern; but we get the 99th percentile of the variable itself  $U_{99}$ . In P10, table 2, Equation 1 has been modified as the "c" was removed:

$$U_{99} = \sqrt{u_0^2 + v_0^2}$$

P8, Table 2, Eq.(2): Uc should be Uc,99. [In Re-expression this is indeed done but not here in Fr expression]

We agree with the reviewer's concern, but in this equation we do not need the 99th percentile of U. Instead, we are calculating the 99th percentile of  $Re_{99}$  and  $Fr_{99}$ . In other words: The 99th percentile is calculated from the derived variables and not from the input variables such as U. The Equations 2 and 3 from Table 2 (page 10) are:

$$Fr_{99} = \frac{U}{\sqrt{gh}} \dots (2)$$

$$Re_{99} = \frac{UD}{v}....(3)$$

P8, Table 2, Eq.(4): Substituting this relationship straight into Eq.(7) saves you one unnecessary intermediate quantity (s), equation (Eq.4) and text (line 200) to explain it.

We agree with the reviewer's point of view, P10, Table 2, Eq.(4) was deleted and added in L229:

"Equation 5 is taken from Soulsby and Whitehouse (1997), where s ( $s = \frac{\rho_s}{\rho_w}$ ) represents the specific gravity of sediment grains."

L197-199: Please use equations instead of sentences to present parameter values, for example:  $\rho = 2650 \text{ kg/m}^3$ .

We agree with the reviewer's point of view, L228 – 235 were modified:

"The values assumed for all OWFs sites are:

$$\rho_s = 2650 \, kg/m^3$$
,  $\rho_w = 1027 \, kg/m^3$ ,  $v = 1.3x10^{-6} m^2/s$ ,  $g = 9.8 \, m/s^2$

Where  $\rho_s$  is the sediment density, based on Soulsby (1997).  $\rho_w$  is the water density, v is the kinematic viscosity and g the gravitational acceleration. Equation 4 was calculated based on van Rijn (1984), where  $D_*$  is the non-dimensional grain diameter, and this is first calculated to calculate the critical Shields parameter ( $\theta_{cr}$ ), which represents the initiation of motion at the bed, as proposed by Soulsby and Whitehouse (1997).

Equation 5 is taken from Soulsby and Whitehouse (1997), where s ( $s = {\rho_s}/{\rho_w}$ ) represents the specific gravity of sediment grains. The  $d_{50}$  represents the median sediment grain size.

L198: 1.3E-6 m/s^2: please correct units (should be m^2/s) and avoid computer notation L229 has been updated to:

"
$$\rho_s = 2650 \ kg/m^3$$
,  $\rho_w = 1027 \ kg/m^3$ ,  $v = 1.3x10^{-6}m^2/s$ ,  $g = 9.8 \ m/s^2$ "

L200: Please also introduce D50

L235  $D_{50}$  has been changed to  $d_{50}$  and it was introduced:

"The  $d_{50}$  represents the median sediment grain size."

L212 (and further): Keulegan-Carpenter? wave velocities? Please give explanation, not just symbols.

L2012-216 the Keulegan–Carpenter number  $(KC_{99})$  and the mobility parameter  $(\theta_{99}/\theta_{cr})$  were introduced:

"Additionally, the Keulegan–Carpenter number ( $KC_{99}$ ) was calculated, which is used to determine the relative influence of drag and inertia forces, the formation of vortices, and the potential for sediment transport (Sumer & Fredsøe, 2002). The mobility parameter ( $^{\theta_{99}}/_{\theta_{Cr}}$ ) is considered a key controlling factor for scour, as it reflects the onset of sediment motion under given flow conditions (Soulsby, 1997; Whitehouse et al., 2000). The calculations of those two parameters are shown in Table 2, equation 9 and 20."

L217: There seems to be a change of terminology: Turbine or OWES? It seems that in this piece of text you are using a different name than before. Or am I missing something?

We thank the reviewer for pointing this out. The inconsistency in terminology has been addressed. To ensure clarity and consistency, we have adopted the standardized term "OWES" throughout the manuscript.

L224: "analyzed in more detail": This is a bit puzzling to me. Do you mean: "were included in our analysis"? What did you with the other 220: analyse them "in less detail" or simply discard them?

L275-276 to avoid confusion the sentence was changed and specifically mentioned that 460 OWEs were analyzed in this study:

"As a result, 460 OWES across the nine OWFs were analyzed in this study."

L227: "the next chapter" should be section 2.4? (please be specific, papers do not really have chapters)

L279-280 was modified to:

"A detailed description of this part of the workflow is provided in section 2.4."

P10, Figure 2: This figure is very clear and helpful. Typo in figure: should be "Acquisition"

P13, Figure 2 was updated, the word "acquisition" has been corrected and Uc has been rewritten as U.

L234: Here I welcome the use of "percentile" (rather than quantile). Please check everywhere.

We agree that percentile is the more suitable term. The term "quantile" has been changed to "percentile" throughout the manuscript.

L239: "foodprint" should be "footprint"

L293 "foodprint" was modified to "footprint"

L246: In the spirit of PCA, I think it is most clear to speak of "linear combinations".

L302 to give more clarity, linear combinations was added to the sentence:

"....,which are linear combinations of the original variables...."

L253: I am confused by the number of 692, as I thought that you earlier had only 460 OWES left (see line 224). Or am I missing something here?

L310-311, we have clarified the reason of 692 turbines used in this analysis, we have 460 OWES analyzed from this study and an additional 232 OWES were included from Melling (2015). This makes a total of 692 OWES:

"In this study, the PCA was applied to a dataset of 692 OWES, including 460 from our analysis and an additional 232 OWES from London Array and Thanet OWF, based on Melling's (2015) data"

L273 (as pointed out earlier): "scour depths" should be "relative scour depths"

Throughout the manuscript, we have revised the terminology to consistently use "relative scour depth" wherever this was the intended meaning, replacing the more general term "scour depth" to improve clarity and precision.

L274: This is an incorrect way of present S/D-values: "0.29 S/D" should be "S/D = 0.29" or an "S/D value of 0.29".

L352-355 the values were presented in the way the reviewer has suggested:

"....the values range from S/D=0.29 to S/D=2.49. This OWF is characterized by fine and medium sands. In contrast, the smallest relative scour depth occurred at the OWF of Lynn and Inner Dowsing (Figure 3 3.F), with values from S/D=0.12 to S/D=0.92, which...."

L277: "influence" should be "influenced"

L357 the word was corrected to "influenced"

P13. Figure 3: I see letters A-I rather than numbered markers 1-9. Please adjust/clarify. Also, I think "colormap" should be "colorbar".

P16, Figure 3, the caption was corrected to "Lettered markers (A-I), and colormap was changed to colorbar.

P14, Figure 4: Please state that these angles are presented in degrees and I do not see the added value of presenting so many digits. Suggest rounding off to degrees. I don't think this really is a percentage (number between 0 and 100) but a number between 0 and 1. To my knowledge the value in the right column is the absolute value of the cosine of the angle in the middle column. And why some printed in boldface and others in regular typeface?

Thanks to the reviewer for pointing this out. We have clarified in the caption that the angles are given in degrees. Furthermore, the column label "Correlation" has been specified as the cosine-based correlation with S/D. This metric reflects the strength of the correlation (ranging from 0 to 1), with values closer to 1 indicating stronger correlations. On the other hand, the boldface indicates the variables that have the strongest correlations with S/D. The caption of Figure 4, in page 17 has been modified:

"Figure 4: a) PCA biplot, illustrating the correlation between variables and relative scour depth. b) The table detailing the angles between the relative scour depth and the other variables (in degrees), along with the magnitude cosine-based correlation (values from 0 to 1), where values closer to 1 indicate stronger correlation."

L318 space missing, L321 "flow-related", L323 "in-depth"

L318, L321 and L323 do not longer exist, as the corresponding section was fully rewritten.

L326: why not more directly phrase as "sediment classes" instead of "soil classes"?

Thanks to the reviewer for the suggestion. Our study is following the terminology used by Annad et al. (2021), where the classification based on  $d_{50}$  was named as "soil classes", and we want to maintain consistency with the terminology. As a result, the term "soil classes" is being used in the manuscript.

L336: This adds up to a total of 727 data points. How does this relate to the numbers mentioned earlier?

Thanks to the reviewer for pointing this out. Upon revisiting the dataset, we identified a misleading count in the number of data points. The initial total mistakenly included all turbine grain size values, regardless of whether corresponding relative scour depth data was available. We have corrected this by considering only the  $d_{50}$  values for OWEs with available S/D data. As a result, the total number of data points used in the analysis is now 692. This correction has been reflected in the revised manuscript, specifically in L426–429:

"After clustering, six soil classes were obtained: cohesive sediment (  $d_{50} \le 63 \mu m$ ) with 5 data points, fine sand ( $63 \le d_{50} < 200 \mu m$ ) with 203 data points, medium sand ( $200 \le d_{50} < 630 \mu m$ ) with 249 data points, coarse sand ( $630 \le d_{50} < 2000 \mu m$ ) with 170 data points, fine gravel ( $2000 \le d_{50} < 6300 \mu m$ ) with 18 data points, and medium gravel ( $2000 \le d_{50} < 6300 \mu m$ ) with 49 data points."

L350 another example of "relative scour depths"?

Please refer to the comment above. This has been revised throughout the manuscript.

L359 "depth" (should be singular)

L467 The word depth can be found in singular.

P19, Figure 6: Please be consistent in terminology: replace "normalised scour depth" with "relative scour depth"

Figure 6, page 23 has been updated in the manuscript, and the terminology was simply established as h/D

L413: change "relatively minor" into "small"?

L594 minor has been changed into "small"

L416: change "trends" into "correlations" (the word trend suggests something evolving over time)

L598 the word trend has been changed into "correlation"

L424-425: can be shortened into "disagree"

L608 has been shortened to "disagree"

L426 "depth" (should be singular)

L609 depth has been written as singular

L457-460: you mention the influence of water depth on the way wave height impacts the system. What about the correlation between water depth and current speeds?

L495-498 the correlation between flow velocity and water depths has been mentioned:

"Furthermore, Link and Zanke (2004) observed that maximum relative scour depth tends to develop more slowly and reach lower values in deeper water depth, even under constant average flow velocity, due to reduced shear velocity over the undisturbed bed. This highlights that the relationship between relative water depth and scour is not necessarily linear."

L482, start of Section 3.5: This comes a bit as a surprise since it is not really prepared for in the Introduction. How does Section 3.5 contribute to the goal set out in Section 1? Please also see my earlier suggestion to add a sentence to the Introduction (L99).

L111-115 the introduction for section 3.5 has been included to improve the logical flow and to better prepare the reader to this section:

"This analysis aims to (1) identify universal drivers of scour across all sites, (2) assess sediment specific trends by grain size ( $d_{50}$ ) and (3) evaluate site specific variability at the level of three selected OWFs

(Robin Rigg, Lynn and Inner Dowsing and London Array). The site specific analysis in Section 3.5 assesses the robustness of the global correlations under local conditions and provides insight into how local conditions influence scour behavior""

L552-554: again confusing presentation of S/D-values. Please see my remark on L274.

L834-835 the values were presented in the way the reviewer has suggested:

"...from S/D=0.2 to S/D=2.1. This variability differs markedly from the consistently larger relative scour depth observed at Robin Rigg and the limited maximum depth of up to S/D=1.0..."

L606, Section 4 in its entirety: I am a bit puzzled as to the role/meaning of this section. Is it a discussion, part of it, or rather a summary of all results so far. And how does it then relate to sections 5 and 6 that are still to come? Please reconsider merging the content of sections 4 and 5 - and consider the header "Discussion" as a title? This would avoid unnecessary doubling with the conclusion, which would certainly help me as a reader.

Thanks to the reviewer for this observation. We totally understand the reviewers concerns regarding the role for section 4. To clarify, we have done section 4 to highlight the broader implications of the identified correlations between relative scour depth and site conditions for scour predictions frameworks, rather than summarizing the results. On the other hand, the header of this section has been called "4. Discussion" organized as follows:

- "4. Discussion
- 4.1 Discussion of implications for scour predictions for OWFs
- 4.2 Discussion of limitations and future research"

**L629 Why emphasis on nonlinear?**

L937 we emphasize nonlinear, because scour development often shows threshold-like responses and complex interaction among variables, and those complex interactions are not well captured by linear models.

L648, again please make sure that Keulegan-Carpenter is properly introduced/explained in the manuscript.

L212-213 the Keulegan-Carpenter number ( $KC_{99}$ ) has been introduced/explained:

"Additionally, the Keulegan-Carpenter number ( $KC_{99}$ ) was calculated, which is used to determine the relative influence of drag and inertia forces, the formation of vortices, and the potential for sediment transport (Sumer & Fredsøe, 2002)."

L665: I welcome the structure of the conclusion. Please see how you can incorporate this also in the aim set out in the Introduction, so as to help the reader in what to expect.

L111-113 the structure from the conclusion has been included in the introduction:

"This analysis aims to (1) identify universal drivers of scour across all sites, (2) assess sediment specific trends by grain size ( $d_{50}$ ) and (3) evaluate site specific variability at the level of three selected OWFs (Robin Rigg, Lynn and Inner Dowsing and London Array)."

**References:**

Annad, M., Zourgui, N. H., Lefkir, A., Kibboua, A., and Annad, O.: Scour-dependent seismic fragility curves considering soil-structure interaction and fuzzy damage clustering: A case study of an Algerian RC Bridge with shallow foundations, Ocean Eng., 275, 114157, 2023.

Chiew, Y. M.: Local scour at bridge piers, Publication of: Auckland University, New Zealand, 1984.

Corvaro, S., Mancinelli, A., and Brocchini, M.: Flow dynamics of waves propagating over different permeable beds, Coast. Eng. Proc., 35, 35–35, 2016.

Federal Highway Administration (FHWA): Evaluating scour at bridges (HEC-18, Publication No. FHWA-HIF-12-003), U.S. Department of Transportation, 2012.

Gabriel, K. R.: The biplot graphic display of matrices with application to principal component analysis, Biometrika, 58(3), 453–467, 1971.

Hu, R., Wang, X., Liu, H., and Lu, Y.: Experimental study of local scour around tripod foundation in combined collinear waves—current conditions, J. Mar. Sci. Eng., 9(12), 1373, 2021.

Jolliffe, I. T., and Cadima, J.: Principal component analysis: a review and recent developments, Philos. Trans. R. Soc. A Math. Phys. Eng. Sci., 374(2065), 20150202, 2016.

Link, O., and Zanke, U.: On the time-dependent scour-hole volume evolution at a circular pier in uniform coarse sand, in: Proc. 2nd International Conference on Scour and Erosion (ICSE), 2004.

Matthieu, J., and Raaijmakers, T.: Interaction between offshore pipelines and migrating sand waves, 2012.

Qu, L., An, H., Draper, S., Watson, P., Zhao, M., Harris, J., Whitehouse, R., and Zhang, D.: A review of scour impacting monopiles for offshore wind, Ocean Eng., 301, 117385, 2024.

Roulund, A., Sutherland, J., Todd, D., and Sterner, J.: Parametric equations for Shields parameter and wave orbital velocity in combined current and irregular waves, 2016.

Sheppard, D. M., Zhao, G., and Ontowirjo, B.: Local scour near single piles in steady currents, in: Water Resources Engineering, 1809–1813, ASCE, 1995.

Sheppard, D. M., Ontowirjo, B., and Zhao, G.: Conditions of maximum local scour, in: Proc. Stream Stability and Scour at Highway Bridges, Resources Engineering Conference, 1991–1998, edited by Richardson, E. V. and Lagasse, P. F., ASCE, Reston, VA, 1999.

Smith, J. D., and McLean, S. R.: Spatially averaged flow over a wavy surface, J. Geophys. Res., 82(12), 1735–1746, 1977.

Soulsby, R. L., and Whitehouse, R. J. S.: Threshold of sediment motion in coastal environments, in: Proc. 13th Australasian Coastal and Ocean Engineering Conference and 6th Australasian Port and Harbour Conference (Pacific Coasts and Ports' 97), Christchurch, NZ, Vol. 1, 145–150, Centre for Advanced Engineering, University of Canterbury, 1997.

Sumer, B. M., Petersen, T. U., Locatelli, L., Fredsøe, J., Musumeci, R. E., and Foti, E.: Backfilling of a scour hole around a pile in waves and current, J. Waterw. Port Coast. Ocean Eng., 139(1), 9–23, 2013.

Whitehouse, R. J. S., Harris, J. M., Sutherland, J., and Rees, J.: The nature of scour development and scour protection at onshore wind farm foundations, Mar. Pollut. Bull., 62(1), 73–88, 2011.

Whitehouse, R. J. S., and Harris, J.: Scour prediction offshore and soil erosion testing, 2014.

Zanke, U. C. E., Hsu, T.-W., Roland, A., Link, O., and Diab, R.: Equilibrium scour depths around piles in noncohesive sediments under currents and waves, Coast. Eng., 58(10), 986–991, 2011.

---

## Author Response (AR2)

**Authors' reply to comments**

Preprint wes-2025-41

Title: Scour variability across offshore wind farms (OWFs): Understanding site-specific scour drivers as a step towards assessing potential impacts on the marine environment

Authors: Karen Garcia, Christian Jordan, Gregor Melling, Alexander Schendel, Mario Welzel, Torsten Schlurmann

The authors would like to thank the reviewers again for their constructive and thorough comments and suggestions for our paper. We believe that your feedback has helped us significantly improve the quality of the manuscript. To take into account all the feedback, the paper has been carefully revised. One of the reviewers provided additional comments, we have thoroughly revised and improved the manuscript accordingly. Please find our responses (blue) and revised text blocks (*blue*, *italic*) below the quoted reviewer comments (black).

**Response to Referee #2**

It's been a pleasure to see that, in their revised version of the manuscript, the authors have addressed most of my comments in a satisfactory manner. In my opinion, the manuscript has improved significantly and is in its current form almost ready for publication.

However, a couple of issues appear/remain, possibly resulting from a misunderstanding arising from my previous comments. I think it would be a pity not to clear this up:

- [1] If the dimensionless parameter D/d50 (introduced in the abstract) is supposed to be the relative median grain size, it should be d50/D and not the other way around. Instead you now have defined the relative monopile diameter, i.e. D scaled against the d50. I think this is really important for interpretation, because increasing d50 implies that d50/D increases too (which would be clear) but is reciprocal D/d50 decreases (which is confusing if you call it the relative median grain size). This undesired situation is reflected in Fig.8ab, which show mirrored patterns just because of what is in my opinion an awkward definition of the dimensionless parameter...
- [1] The authors appreciate the reviewer's comment, the parameter originally referred to as the relative median grain size was indeed expressed as D/d50, which corresponds more accurately to the relative monopile diameter. Following the reviewers comment, we have revised the terminology and definition throughout the manuscript to reflect d50/D as the correct form of the relative median grain size. Accordingly, the PCA analysis (Fig. 4) and the plots in Figure 8 have been updated to ensure consistency and correct interpretation:

a)

| Variables                 | $\theta$ to | Cosine-     |
|---------------------------|-------------|-------------|
|                           | S/D         | based       |
|                           |             | Correlation |
|                           |             | with S/D    |
| $d_{50}/D$                | 173         | 0.99        |
| h/D                       | 166         | 0.97        |
| KC 99          | 43          | 0.72        |
| $\theta_{99}/\theta_{cr}$ | 50          | 0.70        |
| Re 99          | 137.        | 0.63        |
| $(^{U}/_{U_{cr}})_{99}$   | 92          | 0.46        |
| Fr 99   | 91          | 0.02        |

Figure 4: a) PCA biplot, illustrating the correlation between variables and relative scour depth.
b) The table detailing the angles between the relative scour depth and the other variables (in degrees), along with the magnitude cosine-based correlation (values from 0 to 1), where values closer to 1 indicates stronger correlation. Boldface highlights the variables with the strongest correlation with relative scour depth.

**L340-358 description of Figure 4 has been updated:**

"As shown in the biplot, PC1 and PC2 account for 73.29% of the variation in the data set. This high percentage indicates that these two components capture most of the significant patterns in the data, allowing for a meaningful interpretation of the relationships among the variables. In the biplot, each vector stands for a variable, with the direction and magnitude of the vector reflecting its contribution to the principal components. The variables that contribute the most to the variance in PC1 are the mobility parameter, the Froude number, and Keulegan Carpenter number, with shares of 0.5, 0.4, and 0.3, respectively. In contrast, the variance in PC2 is primarily explained by the pile Reynolds number, the relative water depth and the Froude number, with shares of 0.7, -0.4, and 0.3, respectively. This significant contribution of the mobility parameter, the Froude number, and the Keulegan Carpenter number to PC1 suggests that variations in these hydrodynamic parameters are critical in shaping the principal dynamics of the dataset. The table (Fig. 4b) next to the biplot provides further insight by showing the angular distances between the S/D vector and each of the other variables, as well as their respective correlation coefficients. One of the key observations is that the relative scour

depth has the strongest negative correlation of 0.99 with the relative grain size. This highlights the critical influence of sediment size on scouring processes, even though it does not account for much of the variance captured by the first two principal components. This observation can be explained by the underlying physical processes that affect scour depths. As noted by Whitehouse (2010) for non-cohesive sediments, larger sediment sizes are more resistant to erosion, resulting in reduced scour depths. Therefore, while relative grain size is strongly correlated with scour depths, it does not explain the broader variability in the data that is influenced by other factors. The next strongest correlation is with the relative water depth with a correlation factor of 0.97,"

Figure 8 page 25 has also been updated:

Figure 8: Relative scour depth against (a) the relative grain size, and (b) grain size. The solid red curves represent the rational polynomial line fits to the 99th percentile of relative scour depth, for various relative grain size and grain size .Data points for London Array and Thanet OWFs are included from Melling (2015).

L547-560 description of Figure 8 has also been updated:

"Figure 8a summarizes the findings from the PCA analysis (Figure 4) by plotting the relationship between the relative scour depth and relative grain size across all the sampled locations. Figure 8b is also shown here to support figure 8a by representing the data in terms of the grain size, allowing the comparison of dimensional and non-dimensional relative grain size. Figures 8a and 8b illustrate a discernible correlation where the largest relative scour depth occurs predominantly in fine to medium sands, as indicated by the rational polynomial line which approximates the upper limit of relative scour depth for various relative grain size (Figure 8a) and grain size (Figure 8b). Similar to the correlation presented in Figure 7, this curve was derived by fitting a rational polynomial function to the 99th percentile values of relative scour depth, computed within uniform interval of relative grain size (e.g., 0.00001) and grain size (e.g., 25 µm). The correlation shown in figures 8a and 8b are well explained."

[2] I still disagree with how you define percentiles in Eqs.(1)-(3). Mathematically speaking, adding a subscript 99 on the left-hand side of a definition does not mean that you take the 99-th percentile of whatever is written on the right-hand side too. In other words, by writing it this way you suggest that the  $U_99$ , which is supposed to be the 99th percentile, equals something that varies over time, namely  $sqrt(u0^2+v0^2)$ . There are two easy ways to fix this: o First plainly define the Froude number, i.e. Fr = U/sqrt(gh) and then state you will be using the 99th percentile of that quantity, for which you use the symbol  $Fr_990$  Combine everything in a single expression by using a subscript (and brackets) also on the right-hand side:  $Fr_99 = U/sqrt(gh)$ \_99

[3] N.B.: Same for Fr\_99 in Eq.(2) and Re\_99 in Eq.(3). Similarly, I think the tau\_max in Eq.(19) should be tau\_max99

[2] and [3] We would like to thank the reviewer's clarification. We now understand the confusion caused by our notation. Following the reviewer's suggestion, we have decided to combine everything in a single expression using subscripts (and brackets) on the right side. Equations 1–3, 9 and 19 have been updated in Table 2 on page 10 and 11:

| Variable                       | Equation                                          |     |
|--------------------------------|---------------------------------------------------|-----|
| Velocity magnitude             | $U_{99} = (\sqrt{u_0^2 + v_0^2})_{99}$            | (1) |
| Froude number                  | $Fr_{99} = \left(\frac{U}{\sqrt{gh}}\right)_{99}$ | (2) |
| Pile Reynolds number           | $Re_{99} = \left(\frac{UD}{v}\right)_{99}$        | (3) |
| Keulegan-Carpenter number (KC) | $KC_{99} = \left(\frac{U_m T_p}{D}\right)_{99}$   | (9) |

Shields parameter (
$$\theta$$
)
$$\theta_{99} = \left(\frac{\tau_{max}}{(\rho_s - \rho_w)gd_{50}}\right)_{99}$$
 (19)

[4] Terminology: a velocity is a component of the flow vector that can be positive or negative, but what you introduce in Eq.(1) is no longer a velocity: it is a velocity magnitude (also known as speed).

[4] Thanks to the reviewer for pointing this out, the name of velocity has been change to velocity magnitude in Table 2, equation 1, page 10:

"Velocity magnitude:
$$U_{99} = (\sqrt{u_0^2 + v_0^2})_{99} \dots (1)$$
"

[5] The caption of Figure 4 still does not seem to explain explicitly what boldface means. Apart from that, I still do not understand the benefit of presenting the angles with as many as two decimals. Same remark for the percentages used in Fig.4 and the main text. Moreover, the spelling correction from EMODET to EMODNET has not been carried out consistently throughout the entire manuscript. Also on other occasions spelling/grammar is sloppy, please check this carefully. Further, I still notice unclear paragraph breaks, sometimes with white vertical spacing, and in other cases with a 'hard' return. Please check and correct for consistency. And finally, even though the authors said to agree with my earlier suggestion to replace "trend" with "correlation" (since no time element is involved), in the newly added texts they continue to use "trend" in this context on many occasions (L405 L406 L628 L654 L656 L659 L661 L755 as well as captions of FIGURES 7 & 8 of the track-changes manuscript). Please reconsider.

[5] Thanks to the reviewer for this detailed feedback and have carefully addressed each point as follows:

• The figure 4 caption has been updated and contain the explicitly explanation of the Boldface, and can be found in Page 17, L339:

"Boldface highlights the variables with the strongest correlation with relative scour depth."

- The angles decimals has been reduced in the table and in the text, see changes in answer of comment 1.
- We have corrected carefully the spelling correction from EMODET to "EMODNET".
   The spelling/Grammar has been checked carefully, and the paragraphs breaks have been added where was need in the whole manuscript.
- The word "trend" has been change with "correlation" in L110, L454, L555, L597, L560,
   L603, L653, L694, L704, L706, L707, Figures 8 and 9 do not have the word correlation because they have been added in the text description.

Also, I have some additional remarks listed further below (in which I refer to the line numbers of the track-changes manuscript posted by the authors. Overall, my recommendation is now to publish with minor revision, in which I refer to the remaining issues above and the list of additional remarks further below.

Anonymous, 9 July 2025

**MINOR POINTS**

LINE 37 (line numbers of the track-changes manuscript): Principal Component Analysis (use caps just like in main text)

L35-36 Principal Componen5 Analysis has been changed like in the main text.

LINE 106: Terminology, is it ocean or sea-related? I'd expect sea regimes or marine regimes, but I am not aware of the conventions within the various disciplines.

We appreciate the reviewer's observation, in L104 the terminology has been changed to "marine regimes"

This term is more commonly used in the context of offshore scour research.

LINE 195: depth-averaged

L186 depth averaged was corrected

LINE 209: The (capital)

L199 "the" is has not been change because it continue a sentence:

"On the other hand, the Reynolds number"

LINE 214: This phrasing is not fully accurate in my opinion. It considers the shear stress relative to the onset of sediment motion under given flow conditions.

L204-L205 this paraphrase has been changed to avoid confusions:

The mobility parameter  $({}^{\theta_{99}}/_{\theta_{cr}})$  is considered a key controlling factor for scour, as it represents the degree to which the bed shear stress exceeds the critical threshold for sediment motion under given flow conditions (Soulsby, 1997; Whitehouse et al., 2000).

TABLE 2: "amplitud" should be "amplitude", "angular difference" better replaced by "angle"?

In table 2 those words has been changed and now expressed as the reviewer recommended:

"Amplitude of wave orbital motion at the bed (*A*)"

"Angle between the direction of the wave and the current ( $\alpha$ ")

[LINE 228: With my previous remark, I did not mean to say "displayed" equations (as in centered on a separate line), but "in-text" equations are also fine. It was more about using the equality sign to present parameter values (rather than verbs).]

Thanks to the reviewer for pointing this out, L217 – L219 has been updated:

The values assumed for all OWFs sites are:  $\rho_s = 2650 \text{ kg/m}^3$  (sediment density, based on Soulsby, 1997),  $\rho_w = 1027 \text{ kg/m}^3$  (water density),  $v = 1.3x10^{-6}m^2/s$  (kinematic viscosity),  $g = 9.8 \text{ m/s}^2$  (gravitational acceleration).

LINE 230: make sure to use the Greek symbol 'nu' and not the letter 'v'

L218 the Greek symbol nu has been updated:

```
"v = 1.3x10^{-6}m^2/s (kinematic viscosity),"
```

FIGURE 3: 'Letterred markers A-I' should be 'Lettered markers A-I', or even better, simply 'Letters A-I' or 'The labels A-I'.

The caption of Figure 3 has been updated to "Letter (A-I)"

L552 "shows"

L452 "shows" was added

FIGURES 7 & 8: What does "red rational polynomial line" mean? Don't you just mean "solid red curve" which perhaps represents a "rational polynomial" of a certain mathematical form? If you wish to better explain the nature of the curve, this should be done in the main text.

L515-518 the description of the red rational polynomial line for Figure 7 was added:

"The solid red curve shows a correlation between relative scour depths across all relative water depths, independent of sediment class. It was derived by fitting a rational polynomial function to the 99th percentile values of relative scour depth, computed within uniform relative water depth intervals (e.g., 0.1)."

L558-560 the description of the red rational polynomial line for Figure 8 was added:

"Similar to the correlation presented in Figure 7, this curve approximate upper limit of S/D and it was derived by fitting a rational polynomial function to the 99th percentile values of relative scour depth, computed within uniform interval of relative grain size (e.g., 0.00001) and grain size (e.g.,  $25 \mu m$ )."

L626 "differently colored points" is better replaced with "different markers (colour and shape)"

L514-L516 "differently colored points" was replaced with *different markers* (colour and shape)"

L645 full stop missing

L537 full stop was added

L898 L949 Perhaps a matter of taste, but since the section is already entitled "Discussion", there is no need to repeat "Discussion of ..." in each subsection.

L741 and L776 the titles were modified:

4.1 Implications for scour predictions for OWFs

4.2 Limitations and future research

L906 full stop before citation must be omitted

L748 the full stop was omitted

L958 comma should be omitted

L785 comma was omitted

L959 "the accuracy of these parameters \*is\* limited"

L786 "is" was updated in this line